**Cite this article:** van Leeuwen B, Smeets P, Bovet J, Nave G, Stieglitz J, Whitehouse A. 2020 Do sex hormones at birth predict later-life economic preferences? Evidence from a pregnancy birth cohort study. *Proc. R. Soc. B* **287**: 20201756.

behaviour

hormones, economic preferences, testosterone, developmental origins

**Author for correspondence:**
Boris van Leeuwen
e-mail: b.vanleeuwen@uvt.nl

†These authors contributed equally to this work.

# Do sex hormones at birth predict later-life economic preferences? Evidence from a pregnancy birth cohort study

Boris van Leeuwen[1,†], Paul Smeets[2,†], Jeanne Bovet[3], Gideon Nave[4], Jonathan Stieglitz[5] and Andrew Whitehouse[6]

[1]Department of Economics, Tilburg University, Tilburg, The Netherlands
[2]Department of Finance, Maastricht University, Maastricht, The Netherlands
[3]Department of Psychology, Northumbria University, Newcastle upon Tyne, UK
[4]Department of Marketing, The Wharton School, University of Pennsylvania, Philadelphia, PA, USA
[5]Institute for Advanced Study in Toulouse, University of Toulouse 1 Capitole, Toulouse, France
[6]Telethon Kids Institute, University of Western Australia, Crawley, Western Australia, Australia

(iD) BvL, 0000-0001-7246-2130

Economic preferences may be shaped by exposure to sex hormones around birth. Prior studies of economic preferences and numerous other phenotypic characteristics use digit ratios (2D : 4D), a purported proxy for prenatal testosterone exposure, whose validity has recently been questioned. We use *direct* measures of neonatal sex hormones (testosterone and oestrogen), measured from umbilical cord blood ($n = 200$) to investigate their association with later-life economic preferences (risk preferences, competitiveness, time preferences and social preferences) in an Australian cohort (Raine Study Gen2). We find no significant associations between testosterone at birth and preferences, except for competitiveness, where the effect runs opposite to the expected direction. Point estimates are between 0.05–0.09 percentage points (pp) and 0.003–0.14 s.d. We similarly find no significant associations between 2D : 4D and preferences ($n = 533$, point estimates 0.003–0.02 pp and 0.001–0.06 s.d.). Our sample size allows detecting effects larger than 0.11 pp or 0.22 s.d. for testosterone at birth, and 0.07 pp or 0.14 s.d. for 2D : 4D ($\alpha = 0.05$ and power = 0.90). Equivalence tests show that most effects are unlikely to be larger than these bounds. Our results suggest a reinterpretation of prior findings relating 2D : 4D to economic preferences, and highlight the importance of future large-sample studies that permit detection of small effects.

## 1. Introduction

One of the oldest questions in the social sciences is how people's personality and preferences develop over the lifespan [1]. Why do some people take risks, while others avoid them? What makes some individuals more prosocial than others? Recent studies in economics provide causal evidence that preferences are shaped by environmental factors in childhood [2,3]. Even before birth, the fetal environment can have long-lasting effects on later-life outcomes. In the health and psychological sciences, there is now significant evidence for the developmental origins of health and disease hypothesis, whereby exposure to environmental factors during critical periods of fetal development has life-long influences on health, behaviour and cognition [4], as well as important socio-economic outcomes [5,6]. One frequently studied factor of the fetal environment is the exposure to sex hormones. Exposure *in utero* to sex hormones such as testosterone is thought to have long-lasting organizational effects on the brain [7,8]. Differences in sex hormone exposure may help explain sex differences and within-sex heterogeneity in personality and preferences.

In this paper, we focus on economic preferences. Economic preferences—such as risk preferences, competitiveness, patience and social preferences—are the building blocks of economic models. While we label these preferences as economic preferences, they are clearly relevant for other domains of human decision-making. Economic preferences are related to many decisions in daily life [9,10]. Risk preferences influence health behaviour, leisure activities and financial decisions [11]. Competitiveness is associated with study choices and career paths [12]. Social preferences predict whether people sacrifice resources to contribute to social welfare [13] and predict socially responsible investment behaviour [14].

A large literature links prenatal sex hormones with economic preferences. While several studies report significant relationships between prenatal testosterone exposure and risk preferences [15–17], time preferences [18], trust [19] and social preferences [20], other studies find null effects and conflicting evidence (for reviews, see Apicella *et al.* [21], Parslow *et al.* [22] and Neyse *et al.* [23]). The role that hormones play in the development of preferences is, therefore, not well understood.

The vast majority of previous studies use adult digit ratios (2D : 4D) as a proxy for prenatal androgen and oestrogen exposure. 2D : 4D is the ratio of the length of the second digit to the length of the fourth digit. It is, therefore, minimally invasive and easy to measure. Researchers introduced 2D : 4D as a possible proxy for prenatal testosterone based on the observation that adult men have relatively longer fourth digits than adult women [24]. Subsequent studies suggested that digit ratios are influenced by testosterone exposure during pregnancy [25–28]. Since then, the number of studies relating 2D : 4D to diverse outcomes has exceeded 1400 as of 2019 [29]. Outcomes include aggression, sexual orientation, sports performance, risk taking and prosocial behaviour.

Yet the validity of 2D : 4D as a proxy for prenatal testosterone exposure has recently been called into question [29,30]. While studies using amniotic fluid sometimes find correlations between testosterone levels and 2D : 4D, sample sizes are small and results are inconsistent [25,28]. Digit ratios change after birth [31–33], contradicting the initial hypothesis that these ratios are fixed prenatally. Moreover, a study of individuals with complete androgen insensitivity syndrome found that 2D : 4D variance was similar to controls, suggesting sources of variability that cannot be attributed to androgenic influences alone [34]. Studies among Hadza hunter–gatherers of Tanzania show that purported sex differences in 2D : 4D are not universal [35]. There is also ongoing debate as to whether 2D : 4D sexual differentiation is the product of an allometric shift in shape rather than androgenic influences *per se* [36,37]. A recent study found no correlation between 2D : 4D and testosterone measured at birth from umbilical cord blood [38]. Clearly, studies using other hormonal measures are needed.

An ideal study design would relate direct measures of fetal sex hormones to later-life preferences. But sampling human fetal tissue or experimentally manipulating fetal hormones is not feasible for obvious ethical reasons. Researchers instead must use proxies for fetal hormone concentrations during pregnancy. Currently, there is no 'gold standard' hormonal measure [39]. One possibility is to measure hormone concentrations in amniotic fluid, which requires amniocentesis. Yet amniocentesis is a high-risk procedure that is carried out only when there is a clinical justification, which would introduce selection bias.

Our study, therefore, uses direct measures of neonatal sex hormones collected at birth from umbilical cord blood. Umbilical cord blood is thought to reflect hormone exposure during late gestation [39]. Robust sex differences in cord blood hormone measures have been found [40,41]. It is possible that critical periods for hormone exposure and neurodevelopment occur earlier in gestation. While causal evidence for humans is lacking, nonhuman animal models suggest that separate critical periods exist for genital and neurodevelopmental effects of hormone exposure. Importantly, late gestation was found to be a critical period for neurodevelopment [42,43]. Umbilical cord blood measures can thus provide important new insights.

Our study has two main objectives. First, we link *direct* measures of sex hormones at birth to later-life economic preferences ($200 \leq n \leq 217$, depending on the measure). Second, we test the robustness of earlier findings regarding the relationship between 2D : 4D and economic preferences, but with a larger sample size ($533 \leq n \leq 597$, depending on the measure) than most previous studies. All main analyses were pre-registered (see https://osf.io/xt8s6/?view_only= eb37d6b404e94fd3b9c8952424d588f3). We collected data as part of the ongoing Raine Study, a large and well-studied Australian cohort [44]. Between 1989 and 1991, 2900 pregnant women from Perth, Western Australia, volunteered to participate in the Raine Study. In the present study, we conducted incentivized experiments measuring the economic preferences of the offspring (Raine Study Gen2) when they were 25–27 years old. We consider a diverse array of economic preferences, namely those related to risk tolerance, competitiveness (willingness to compete), time preferences (patience), social preferences (positive and negative reciprocity, i.e. responding to a kind action with a kind action and responding to an unkind action with an unkind action, respectively) and also trust and dishonesty.

## 2. Material and methods

### (a) Participants

Participants were recruited from the Raine Study. The Raine Study consecutively recruited pregnant women and their offspring from King Edward Memorial Hospital (KEMH) in Perth or nearby private practices between 1989 and 1991. Inclusion criteria were gestational age between 16 and 20 weeks, English language skills sufficient to understand the implications of participation, an expectation to deliver at KEMH and an intention to remain in Western Australia to facilitate future follow-ups. Ninety per cent of eligible women agreed to participate in the initial wave [45]. The offspring and families have participated in regular follow-up assessments [44]. At the time of the present study, the offspring (Raine Study Gen2; participants for the current study) were between 25 and 27 years old. The cohort design of the Raine Study allows us to combine existing measures (in particular sex hormones measured at birth and 2D : 4D measured later in life) with economic preferences measured in our experiments.

All active Raine Study participants within the cohort were invited to take part in the present study. A total of 1500 participants were invited, of which 742 (448 women, 294 men) participated, of which 661 completed all experiments and survey questions. The experiment was conducted online, and participants received a link via email with their personal login details. All participants provided informed consent at the beginning of the study. The study was pre-registered (https://osf.io/ xt8s6/?view_only=eb37d6b404e94fd3b9c8952424d588f3)       and

ethics approval was obtained from the Human Research Ethics Office of the University of Western Australia.

## (b) Experimental and survey measures

The experiment consisted of five incentivized tasks and several survey questions. Risk attitudes were measured both using the incentivized method of Eckel & Grossman [46] and a self-reported measure based on Dohmen et al. [11]. To measure competitiveness, we used a version of the task developed by Niederle & Vesterlund [47], adapted for online use by Buser et al. [48]. Patience (time preferences) was measured using a validated survey question from the Global Preferences Survey by Falk et al. [10]. To measure trust and positive reciprocity, we used a version of the Trust Game [49] and to measure negative reciprocity we used a version of the Ultimatum Game [50]. Dishonesty was measured using a coin-flipping task [51]. Details on the elicitation methods are provided in the electronic supplementary material, S1.

To provide real financial incentives, 10% of study participants were randomly selected for payment (after the study was closed). Those who received payment earned between 20 AUD (14 US$) and 525 AUD (354 US$) (mean 219 AUD, s.d. 110 AUD). Complete experimental instructions and survey questions can be found in the electronic supplementary material, S2. Most participants took approximately 40 min to complete the study.

## (c) Hormone measures, digit ratios and sex

For a random subsample (30% of the initial Raine Study sample), mixed arterial-venous umbilical cord blood was obtained at birth. Cord blood samples were collected immediately after delivery and frozen and stored. In January 2010, blood samples were analysed using mass spectrometry to assess total testosterone and other androgens and oestrogens (for details see Keelan et al. [41] and Hickey et al. [52]). We also use measures of bioavailable testosterone (BioT) and bioavailable oestradiol (BioE2), which we derive from other measured hormones. Bioavailability represents the fraction of total testosterone or oestradiol either free or bound to serum albumin. It was calculated using the formulas: $BioT = free\ testosterone + albumin\text{-}bound\ testosterone$ [41] and $BioE2 = free\ oestradiol + albumin\text{-}bound\ oestradiol$ [38]. Albumin levels were adjusted using published reference values to take into account the decrease in serum albumin concentrations with gestational age [53].

Digit ratios were measured between 19 and 22 years of age. The length of the second and fourth digit was measured for both hands from hand photocopies using Vernier callipers, following the procedures of Caswell & Manning [54].

We characterize study participants as men or women based on the biological sex that was determined at birth.

## (d) Other measures

We conduct our analyses with and without control variables. Fetal growth (i.e. birth weight and gestational age) is related to sex hormone measures at birth [41]. Socio-economic status (SES) of parents has been related to the economic preference measures of their children (e.g. [11,55]). We thus control for family SES (i.e. family income and parental education) and two other parental measures (i.e. whether biological parents are co-resident, and whether they were born in Australia or elsewhere). These data were recorded in surveys administered during pregnancy. Previous studies have suggested that sex differences in 2D : 4D may not be universal [35] and find differential associations between 2D : 4D and economic preferences depending on ethnicity [17]. We, therefore, also conduct analyses by self-reported ethnicity of the parents. Electronic supplementary material, table S2 lists summary statistics for all control variables.

## (e) Statistical analysis and pre-analysis plan

We pre-registered the main analyses and variable definitions. An important advantage of pre-registering for our study is that it restricts the degrees of freedom of researchers and precludes undisclosed flexibility in data analysis. The pre-analysis plan and the full experimental and survey design can be found at https://osf.io/xt8s6/?view_only=eb37d6b404e94fd3b9c895 2424d588f3.

In the pre-analysis plan, we pre-registered 4 × 2 studies. We planned to study the relationship between neonatal hormones, stress and four types of outcome measures: (i) risk preferences, (ii) social preferences, (iii) competitiveness, and (iv) financial decision-making. In this paper, we report on the studies relating to neonatal hormones and risk preferences, social preferences and competitiveness. We added patience (from the financial decision-making part), as other papers have considered patience along with the other preferences as well.

We pre-registered four hormonal measures as the main independent variables of interest: total testosterone, BioT, the ratio between total testosterone and total oestradiol (AE ratio), and the ratio between BioT and BioE2 (BioAE ratio). Hence, our pre-registered analyses are the 4 × 8 associations between the four hormone measures (total testosterone, BioT, AE ratio, BioAE ratio) and eight preference measures (incentivized risk tolerance, self-reported risk tolerance, competitiveness, patience, trust, positive reciprocity, negative reciprocity, dishonesty). In addition to these pre-registered analyses, we also report associations between 2D : 4D and preferences. We did not perform any ex-ante power calculations, as we aimed to maximize our power by inviting all active participants in the Raine Study.

In the pre-registration, we stated that the main analyses would be ordinary least squares (OLS) regressions of the respective preferences measure on the respective hormone measure, a sex dummy and an interaction term between the hormone measure and the sex dummy. These analyses are reported in the electronic supplementary material. For expositional purposes, we deviate from the pre-analysis plan in the main text by reporting analyses without the sex * hormone interaction term, as well as associations between hormones and preferences by sex. Conclusions from results reported in the main text and electronic supplementary material are identical.

In the regressions, we include all available data. This means that sample sizes may vary between regressions, because of missing data on some variables. All p-values reported are based on OLS regressions (unless stated otherwise) and are not corrected for multiple hypotheses testing.

# 3. Results

## (a) Descriptive statistics

We obtained total testosterone measures for 217 participants (125 women), right hand 2D : 4D measures for 597 participants (350 women) and left hand 2D : 4D measures for 595 participants (349 women). Electronic supplementary material, figure S1 shows kernel density plots of hormone measures at birth and 2D : 4D by sex. For our pre-registered hormone measures (total testosterone, BioT, AE ratio and BioAE ratio), as well as for 2D : 4D, we observe significant sex differences (Kolmogorov–Smirnov tests, all $p < 0.001$). Electronic supplementary material, table S1 shows correlation coefficients for all hormone and 2D : 4D measures. As previously reported, hormonal measures at birth in the Raine Study sample and 2D : 4D are generally uncorrelated [38]. We replicate these findings in our current (sub)sample.

*Proc. R. Soc. B* **287**: 20201756

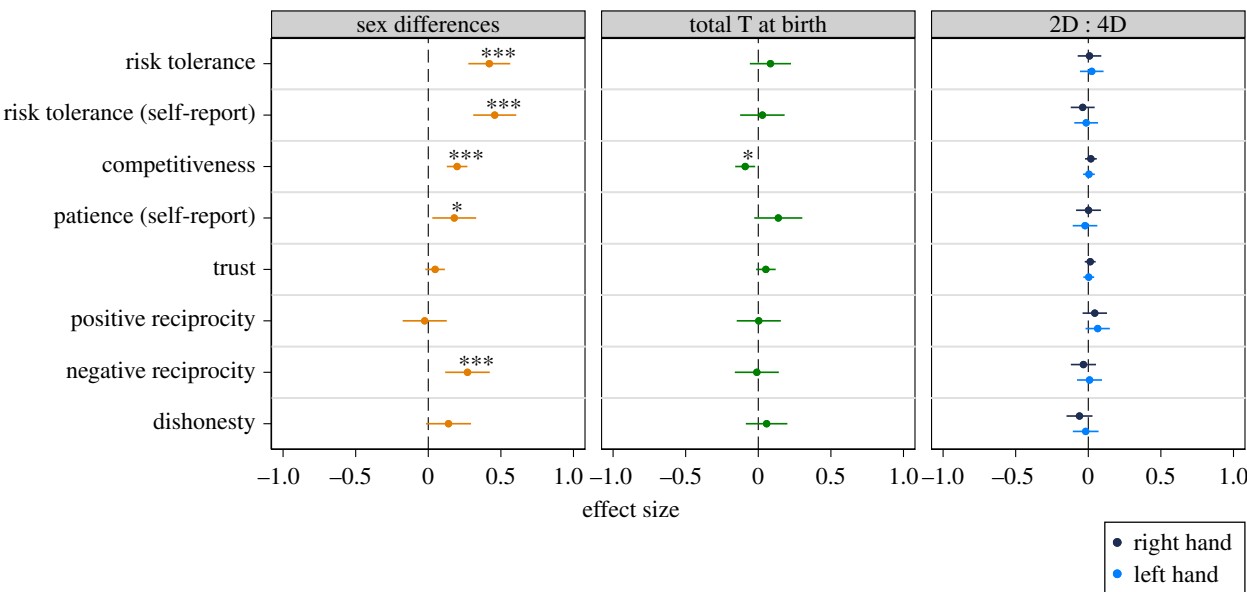

**Figure 1.** Sex differences, testosterone (T) at birth, 2D : 4D and economic preferences. Point estimates and 95% confidence intervals from OLS regressions. The respective preference measure is regressed on a dummy that equals 1 for men and 0 for women (left panel), on T at birth and a sex dummy (middle panel) or the left or right hand 2D : 4D ratio and a sex dummy (right panel). $*p < 0.05$, $**p < 0.01$, $***p < 0.005$. Sex, competitiveness and trust are binary measures, all other measures (including T and 2D : 4D) are standardized to have mean 0 and standard deviation 1. Sample sizes: $n \geq 661$ (sex differences); $n \geq 200$ (total T at birth); $n \geq 533$ (2D : 4D). Regression estimates can also be found in the electronic supplementary material, table S3 for sex differences and T at birth and in the electronic supplementary material, table S4 for 2D : 4D. (Online version in colour.)

The left panel in figure 1 shows that we replicate typically observed sex differences: we find that men are more risk tolerant, competitive, patient and negatively reciprocal than women. We do not observe statistically significant sex differences for trust, positive reciprocity and dishonesty. More details can be found in the electronic supplementary material, S3.

## (b) Testosterone at birth and economic preferences

The middle panel in figure 1 shows associations between total testosterone at birth and preferences. Each estimate controls for sex. For ease of comparison and interpretation, all non-binary outcome measures (both measures of risk tolerance, patience, positive and negative reciprocity, dishonesty) are standardized to have mean 0 and s.d. 1. Electronic supplementary material, table S3 shows the regression estimates.

We find no significant associations between total testosterone at birth and preferences, except for competitiveness. However, this effect runs opposite to the expected direction. A 1 s.d. increase in testosterone at birth is associated with a 9 percentage points (PP) lower likelihood of choosing the competitive tournament payment scheme ($b = -0.090$, $t_{202} = -2.58$, 95% confidence interval (CI) = $(-0.159, -0.021)$, $n = 205$, $p = 0.011$). For all other preference measures, point estimates are small and nonsignificant. Point estimates for risk tolerance (incentivized measure: $b = 0.084$, $t_{214} = 1.17$, 95% CI = $(-0.058, 0.225)$, $n = 217$, $p = 0.245$; self-reported measure: $b = 0.028$, $t_{205} = 0.36$, 95% CI = $(-0.125, 0.181)$, $n = 208$, $p = 0.716$) and patience ($b = 0.138$, $t_{205} = 1.64$, 95% CI = $(-0.027, 0.303)$, $n = 208$, $p = 0.102$) are in the expected direction but nonsignificant. For trust ($b = 0.052$, $t_{205} = 1.52$, 95% CI = $(-0.015, 0.120)$, $n = 208$, $p = 0.129$), positive reciprocity ($b = 0.003$, $t_{205} = 0.04$, 95% CI = $(-0.148, 0.155)$, $n = 208$, $p = 0.964$), negative reciprocity ($b = -0.010$, $t_{197} = -0.13$, 95% CI = $(-0.162, 0.142)$, $n = 200$, $p = 0.896$) and dishonesty ($b = 0.057$, $t_{197} = 0.79$, 95% CI = $(-0.086, 0.200)$, $n = 200$, $p = 0.430$), point estimates are also small and nonsignificant.

Electronic supplementary material, figure S3 shows associations between other hormonal measures that we pre-registered (BioT, AE ratio and BioAE ratio) and preferences, again controlling for sex. These results are comparable to total testosterone at birth. We find no robust associations between any of these hormonal measures and preferences. Out of 24 associations we only find a significant (negative) association between BioT and competitiveness ($b = -0.070$, $t_{201} = -2.11$, 95% CI = $(-0.135, -0.005)$, $n = 204$, $p = 0.036$), and a significant (positive) association between the AE ratio and trust ($b = 0.065$, $t_{205} = 1.98$, 95% CI = $(0.000, 0.130)$, $n = 208$, $p = 0.049$).

We conduct several (pre-registered) analyses to corroborate the robustness of the null effects reported above. Figure 2 shows similar null effects by sex. The left panel of figure 2 shows that the significant (negative) association between testosterone at birth and competitiveness is driven by men, as the point estimate is very close to zero for women. All other estimates are not significantly different from zero. Adding interaction terms between sex and testosterone at birth (which we pre-registered) leads to the same conclusions (see the electronic supplementary material, table S3). Electronic supplementary material, figure S4 shows the associations for BioT, the AE ratio and the BioAE ratio by sex. For men, we find a significant (negative) association between BioT and competitiveness and a significant (positive) association between the AE ratio and trust. All other 46 associations, for either sex, are nonsignificant.

Further (not pre-registered) robustness tests are discussed in the electronic supplementary material, S3. The inclusion of additional controls for fetal growth and SES indicators has little effect on the estimates. Similarly, focusing on a subsample of participants with two Caucasian parents leads to virtually identical results. Allowing for nonlinear effects of testosterone at birth does not affect results. Also for oestrogen measures (oestradiol (E2) and BioE2), the associations between oestrogens and preferences are not statistically significant.

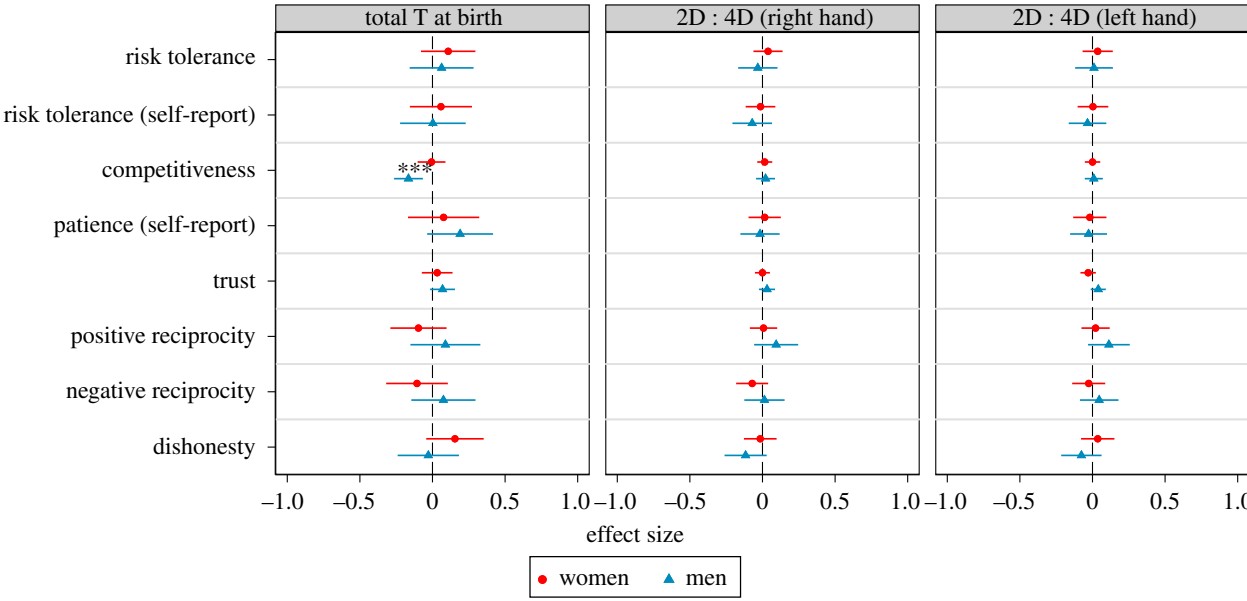

**Figure 2.** Testosterone (T) at birth, 2D : 4D and economic preferences (by sex). Point estimates and 95% confidence intervals from OLS regressions. The respective preference measure is regressed on T at birth (left panel), right hand 2D : 4D (middle panel) or left hand 2D : 4D (left panel), separately for women and men. *$p < 0.05$, **$p < 0.01$, ***$p < 0.005$. Competitiveness and trust are binary measures, all other measures (including T and 2D : 4D) are standardized to have mean 0 and s.d. 1. Sample sizes: $n \geq 112$ (women, total T at birth); $n \geq 88$ (men, total T at birth); $n \geq 303$ (women, 2D : 4D); $n \geq 230$ (men, 2D : 4D). Regression estimates can also be found in the electronic supplementary material, table S3 for T at birth and electronic supplementary material, table S4 for 2D : 4D. (Online version in colour.)

## (c) 2D : 4D and economic preferences

The right panel of figure 1 shows associations between 2D : 4D and preference measures. For both hands, we find null effects: none of the associations are significant and the CIs are tight. Electronic supplementary material, table S4 shows the regression estimates. All point estimates are smaller than 0.07 in absolute value and are not statistically significant (all $n \geq 533$, all $p \geq 0.128$). All 95% CIs lie within values between −0.15 and 0.15, indicating that effect sizes are small.

The middle and right panels of figure 2 show that the null effects for 2D : 4D and preferences hold for both men and women. When testing men and women separately, none of the associations between 2D : 4D and preferences are significantly different from zero. Adding interaction terms between sex and 2D : 4D leads to the same conclusions (see electronic supplementary material, table S4). Further robustness tests are discussed in the electronic supplementary material, S3. Inclusion of additional controls, allowing for nonlinear effects, using average 2D : 4D or looking at subsamples all lead to virtually identical results.

## (d) Power analyses, equivalence tests and multiple testing corrections

Thus far, our results generally indicate an absence of statistically significant associations between sex hormones at birth, 2D : 4D and preferences. Following the suggestion of the reviewers, and beyond the analyses that we pre-registered, we further investigate what effect sizes are implausible based on our data. To this end, we conduct equivalence tests. Following Lakens [56], we test whether our observed effect sizes are smaller than the smallest effect size we have sufficient power to detect. For testosterone at birth and risk tolerance, patience, reciprocity and dishonesty, our sample size of $n \geq 200$ allows us to detect effects larger than 0.22 s.d. units ($n = 200$, $\alpha = 0.05$ and power = 0.90, two sided),

where this effect size is measured as the effect of a 1 s.d. increase in testosterone at birth on the respective preference. For binary preference measures (competitiveness and trust), we have sufficient power to detect effects larger than 0.11 pp. For 2D : 4D, the minimal detectable effect size ($n = 533$, $\alpha = 0.05$ and power = 0.90, two sided) is 0.14 s.d. units (risk tolerance, patience, reciprocity and dishonesty) or 0.07 pp (competitiveness and trust).

To test for equivalence, we use the 'two one-sided tests' procedure (e.g. Lakens [56]). Electronic supplementary material, figure S10 and table S5 show the results. For the association between testosterone at birth and risk attitudes (both measures), trust, positive and negative reciprocity and dishonesty, we can reject effect sizes larger than the minimum detectable effect size. For the associations between testosterone at birth and competitiveness and patience, we cannot reject that the effect is larger than the minimum detectable effect size. For 2D : 4D and preferences, we can reject effect sizes larger than the minimum detectable effect size for all preference measures.

Following our pre-analysis plan, we did not correct for multiple hypothesis testing. We pre-registered 32 tests: eight preference times four hormone measures. Out of these 32 tests, three resulted in a $p$-value smaller than 0.05. We did not pre-register multiple comparison correction, yet see three justifiable corrections for multiple testing, based on 32 comparisons (correct for all tests), eight comparisons (correct for all preference measures across hormones) or four comparisons (correct for all hormonal measures across preferences). In the electronic supplementary material, table S6, we report adjusted $p$-values ($q$-values) based on 32, eight or four comparisons, both using Bonferroni corrections and false discovery rate corrections [57,58]. The (negative) association between total testosterone at birth and competitiveness remains statistically significant if we correct for four comparisons but not if we correct for eight or 32 comparisons. The (negative) association between BioT and

competitiveness and the (positive) association between the AE ratio and trust are both no longer statistically significant, regardless of whether we correct for four, eight or 32 comparisons.

## 4. Discussion

We estimate associations between neonatal sex hormones and later-life economic preferences. Unlike previous research, we use *direct* measures of neonatal sex hormone concentrations, measured from umbilical cord blood collected at birth. We find no significant associations between total testosterone at birth and preferences, except for competitiveness. However, this effect runs opposite to the expected direction. The point estimates are between 0.05 and 0.09 pp for binary preference measures (competitiveness and trust) and between 0.001 and 0.14 s.d. for non-binary preference measures (risk tolerance, patience, positive/negative reciprocity, dishonesty). We also examine the widely used 2D : 4D digit ratio in a sample larger than most previous studies. For 2D : 4D, we estimate null effects with point estimates between 0.003 and 0.02 pp and 0.001 and 0.06 s.d.

In our study, we have hormone measures from umbilical cord blood for over 200 participants (depending on the preference measure), which gives us 90% power to detect effect sizes of 0.22 s.d. ($\alpha = 0.05$, two sided). This means that we lack the power to detect effects smaller than 0.22 s.d. We use a relatively large sample of 2D : 4D measures ($n = 533$), which gives us 90% power to detect effect sizes of 0.14 s.d. ($\alpha = 0.05$, two sided). Based on previous papers, it is plausible that effect sizes are smaller than this. For example, for risk attitudes, Brañas-Garza *et al.* [17] report an effect of 0.08 s.d. ($n = 543$, based on our calculations of results reported in Brañas-Garza *et al.* [59]), Bönte *et al.* [60] report an effect size of 0.10 s.d. ($n = 432$) and Garbarino *et al.* [16] report an effect of 0.16 s.d. ($n = 151$, based on our calculations). To reliably detect such small effects, larger samples are needed. For example, to detect an effect of 0.08 s.d. with 90% power ($\alpha = 0.05$) a sample size of $n = 1600$ is required. This means that previously detected small effects come from low-powered studies, which could lead to overestimation of effect sizes (type-M errors, [61]). It is important to note that many studies also report null effects. Parslow *et al.* [22] survey the literature on 2D : 4D and risk attitudes and find that based on 18 papers, 93 out of 109 reported tests are nonsignificant. Studies with larger samples will have more power to make more definitive claims. It is also interesting to establish relevant moderators for the effects of hormones on outcome measures (see e.g. [62]). For example, one moderator could be levels of adult sex hormones. Moreover, umbilical cord blood measures of sex hormones are relatively new in the literature. We hope that future research will provide additional validation tests of this measure. These validation studies will provide a stronger basis for future studies using umbilical cord blood measures.

Our conclusions have several constraints on generality (cf. [63]). Our sample consists of young Western Australian adults. We believe that our findings would generalize to other adult samples, yet encourage continuous explorations of the effects, particularly in older adults and populations from non-Western societies. In terms of materials, we believe that our findings would generalize to other measurement methods that quantify economic preferences using incentivized economic games or validated survey questions, answered in private. Furthermore, we highlight the importance of measuring sex hormones at birth under the same conditions as stipulated in the paper. We have no reason to believe that the results depend on other characteristics of the participants, materials or context.

Our findings contribute to two streams of literature. First, to the large literature on the relationship between sex hormones around birth and economic preferences, which shows mixed results based on 2D : 4D. These previous studies are inconsistent in the measures used for 2D : 4D. Some studies use the left hand, others the right hand or the average of both hands. Some studies document effects only for men, whereas others only for women [21,22]. Besides 2D : 4D, other measures of testosterone around birth have been used in prior research and all measures have their own limitations. One possibility is to use amniotic fluid collected at mid-gestation. However, this is a risky procedure and therefore typically only conducted with high-risk pregnancies, resulting in selection bias. An alternative is to compare twins with either a male or female co-twin. Cronqvist *et al.* [64] find that women with a male co-twin take significantly more financial investment risk later in life compared to women with a female co-twin. Bütikofer *et al.* [65] find that women with a male co-twin have worse educational outcomes and lower later-life earnings. A limitation of twin studies is that the sex of the co-twin can also affect behaviour through socialization after birth [66]. Turning to our umbilical cord blood measure, it is possible that critical periods for hormone exposure and neurodevelopment occur early in gestation. While different measures of sex hormones around birth all have their own limitations, including ours, it is crucial to find convergent evidence across measures. More than 1400 studies rely on 2D : 4D to assess fetal hormonal exposure, which makes it essential to test the convergent validity of this measure against others.

Second, and more broadly, our work adds to the literature on the origins of preferences. Prior studies reported associations between individual differences in economic preferences and biological factors such as genetics and neuroanatomy (e.g. [67–69]). Moreover, other neonatal factors than hormone exposure have been associated with socio-economic outcomes [5,6,70]. Sex differences in economic preferences are, at least in part, shaped by the social environment [71,72]. Another growing literature shows that early childhood experiences, social interactions and teaching can shape personality and preferences (e.g. [2,3,73,74]). A promising avenue for future research is to shed more light on the interaction between hormones, genetic factors, parenting, social environment and learning in shaping preferences.

Ethics. Ethics approval was obtained from the Human Research Ethics Office of the University of Western Australia.

Data accessibility. The data were collected as part of the Raine Study. No data from the Raine Study can be made available in the public domain as the Raine Study is committed to a high level of confidentiality of the data in line with the informed consent provided by participants. The authors and the Raine Study will assist, however, to provide data access for verification purposes. Interested researchers can apply to access the data through the Raine Study processes.

Authors' contributions. B.v.L. and P.S. initiated the study; B.v.L., P.S., J.B., G.N., J.S. and A.W. designed the study; B.v.L. analysed data; B.v.L. and P.S. wrote the paper; B.v.L., P.S., J.B. and J.S. obtained funding to perform the study. All authors provided comments on the

manuscript and approved its final version for submission. B.v.L. and P.S. contributed equally to this paper.

**Competing interests.** There are no competing interests.

**Funding.** We are grateful to the IAST, Tilburg University and Maastricht University for funding. P.S. acknowledges funding from the Dutch Science Foundation NWO (016.Veni.175.019). G.N. acknowledges funding from the National Science Foundation Early Career Development Program grant (1942917), and thanks Carlos and Rosa de la Cruz for ongoing support. J.S. acknowledges IAST funding from the French National Research Agency ANR (ANR-17-EURE-0010). A.W. is funded by the National Health and Medical Research Council (no. 1077966). The Raine Study has been supported by the National Health and Medical Research Council over the last 30 years, with additional funding for core management provided by the University of Western Australia (UWA), Raine Medical Research Foundation, Telethon Kids Institute, UWA Faculty of Medicine, Dentistry and Health Sciences, Women and Infants Research Foundation, Curtin University, Edith Cowan University, Murdoch University and the University of Notre Dame Australia.

**Acknowledgements.** For excellent research assistance, we thank Dylan Manfredi, Myrte ter Horst and the Raine Study staff. We thank Coren Apicella, Anna Dreber, Alan Fiske, Nordin Hanssen and Jeff Keelan for helpful suggestions and comments. We are extremely grateful to the Raine Study participants who took part in this study and the Raine Study team for cohort coordination and data collection.

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
