## [Reviewer comments · Proceedings of the Royal Society B: Biological Sciences]

Review History

RSPB-2020-1756.R0 (Original submission)

Review form: Reviewer 1

Recommendation

Accept with minor revision (please list in comments)

Scientific importance: Is the manuscript an original and important contribution to its field?

Excellent

General interest: Is the paper of sufficient general interest?

Excellent

Quality of the paper: Is the overall quality of the paper suitable?

Excellent

Is the length of the paper justified?

Yes

Should the paper be seen by a specialist statistical reviewer?

No

Do you have any concerns about statistical analyses in this paper? If so, please specify them explicitly in your report.

No

It is a condition of publication that authors make their supporting data, code and materials available - either as supplementary material or hosted in an external repository. Please rate, if applicable, the supporting data on the following criteria.

Is it accessible?

Yes

Is it clear?

Yes

Is it adequate?

Yes

Do you have any ethical concerns with this paper?

No

Comments to the Author

This is a very interesting paper, resulting from a rigorously executed study on an important topic. I think it's particularly important to publish null results that come from high powered well designed studies like this one, especially when the literature is full of small studies with some "researcher degrees of freedom". The data on hormones at birth used here is really unique and any results are thus interesting. I recommend Accept and just have some minor comments. Addressing these could hopefully make the already very good paper even clearer. My comments appear below in their order of appearance.

Abstract: I think it would be great to know something about the sample size or power already here.

Pre-analysis plan: Is the pre-analysis plan fully followed? It would be informative if the main text could include a clear statement on this. In the SM it says "We pre-registered the main analysis and the way our variables are constructed." Can this be clarified some so that the reader does not need to look at the pre-analysis plan but can understand what tests were pre-specified?

Intro: Perhaps the authors could elaborate somewhat more on why these preferences are interesting to explore, besides in economic models?

Sample sizes: Some sample sizes are mentioned in the intro and some in the methods. Perhaps explain the discrepancies?

Risk measure: Maybe I misunderstood something, but how are participants with multiple switching points treated? (Minor point - Dohmen et al. use a 0-10 scale so perhaps say that your measure is very similar to Dohmen et al?)

Competitiveness measure: Is the competitive tournament performance compared to someone else's part 1 or part 2 performance? If the latter, anyone's or just those who chose to compete? I'm assuming that in cases of ties participants were paid?

Patience: Perhaps include a sentence on how this question has been validated, for those that do not know Falk et al? Are gender differences in patience robust? E.g. for risk preferences I know there is typically a gender difference but with patience I thought it was less clear.

Digit ratios: Just some more info here could be interesting – e.g. were there two raters per hand and what was there correlation? Or just one? Were there any “outliers” and how were these treated?

Figure 2: Is this the average of the two hands for 2D:4D?

Power discussion in the Discussion: In what units is 0.14? What about power and the other null results? Which effect sizes can be ruled out? What is the minimum detectable effect size for the various measures given the data? Or do you think these calculations do not make sense?

Review form: Reviewer 2

Recommendation

Major revision is needed (please make suggestions in comments)

Scientific importance: Is the manuscript an original and important contribution to its field?

Good

General interest: Is the paper of sufficient general interest?

Good

Quality of the paper: Is the overall quality of the paper suitable?

Marginal

Is the length of the paper justified?

No

Should the paper be seen by a specialist statistical reviewer?

Yes

Do you have any concerns about statistical analyses in this paper? If so, please specify them explicitly in your report.

Yes

It is a condition of publication that authors make their supporting data, code and materials available - either as supplementary material or hosted in an external repository. Please rate, if applicable, the supporting data on the following criteria.

Is it accessible?

No

Is it clear?

No

Is it adequate?

No

Do you have any ethical concerns with this paper?

No

Comments to the Author

My review is included in the attached pdf file.

Decision letter (RSPB-2020-1756.R0)

07-Sep-2020

Dear Dr van Leeuwen,

Your manuscript has now been peer reviewed and the reviews have been assessed by an Associate Editor. The reviewers' comments (not including confidential comments to the Editor) and the comments from the Associate Editor are included at the end of this email for your reference. As you will see, the reviewers differ in their opinion of the manuscript, but based on the Associate Editor's evaluation we would like to invite you to revise your manuscript to address the issues raised.

Research ethics:

Use of animals and field studies:

It is a condition of publication that you make available the data and research materials supporting the results in the article. Please see our Data Sharing Policies (<https://royalsociety.org/journals/authors/author-guidelines/#data>). Datasets should be deposited in an appropriate publicly available repository and details of the associated accession number, link or DOI to the datasets must be included in the Data Accessibility section of the

article (<https://royalsociety.org/journals/ethics-policies/data-sharing-mining/>). Reference(s) to datasets should also be included in the reference list of the article with DOIs (where available).

Please submit a copy of your revised paper within three weeks. If we do not hear from you within this time your manuscript will be rejected. If you are unable to meet this deadline please let us know as soon as possible, as we may be able to grant a short extension.

Best wishes,
Professor Loeske Kruuk
Editor
mailto: proceedingsb@royalsociety.org

Associate Editor

Comments to Author:

The manuscript has been reviewed by two experts in the field. Both reviewers like the topic and potential fit for the Journal. I am also generally supportive of the publication of large 'failed-replication' style manuscripts. While Reviewer 1 has only minor suggestions for improvement, Reviewer 2 was quite critical, and thinks that extensive new analyses would be necessary before one could conclude that there were no effects in the data. As the analyses that are currently undertaken are quite simple, I have some sympathy for this position. At the same time, I do understand that the authors might be resistant to this, given that they had a pre-registration, including a pre-analysis plan. I think that a solution might be for the authors to undertake an additional set of analyses and present them in a different section. It would need to be made extremely clear in the MS that these additional analyses were undertaken to satisfy reviewer concerns, and were not part of the pre-registration. It is also more generally important for the

authors to make it extremely clear how closely the pre-registration was followed (see Reviewer 1's comments on this), and to flag any deviation from that pre-registration, no matter how small.

In addition to the comments of both reviewers, I would add that I felt that the structure of the manuscript could be clearer. I would cut the last paragraph of the Introduction, which unnecessarily summarizes the findings – save that for the Discussion. There could usefully be a clearer structure (e.g. aims or objectives, or hypotheses with predictions) laid out at the end of the Introduction that could be used to give the manuscript some explicit structure as subheadings in the Results section. Finally, I would add that details are missing on how individuals were characterized as men or women – by self-report? Please also be consistent in your terminology, I think you should use men and women throughout, not e.g. “females”.

Reviewer(s)' Comments to Author:

Referee: 1

Comments to the Author(s)

This is a very interesting paper, resulting from a rigorously executed study on an important topic. I think it's particularly important to publish null results that come from high powered well designed studies like this one, especially when the literature is full of small studies with some “researcher degrees of freedom”. The data on hormones at birth used here is really unique and any results are thus interesting. I recommend Accept and just have some minor comments. Addressing these could hopefully make the already very good paper even clearer. My comments appear below in their order of appearance.

Abstract: I think it would be great to know something about the sample size or power already here.

Pre-analysis plan: Is the pre-analysis plan fully followed? It would be informative if the main text could include a clear statement on this. In the SM it says “We pre-registered the main analysis and the way our variables are constructed.” Can this be clarified some so that the reader does not need to look at the pre-analysis plan but can understand what tests were pre-specified?

Intro: Perhaps the authors could elaborate somewhat more on why these preferences are interesting to explore, besides in economic models?

Sample sizes: Some sample sizes are mentioned in the intro and some in the methods. Perhaps explain the discrepancies?

Risk measure: Maybe I misunderstood something, but how are participants with multiple switching points treated? (Minor point – Dohmen et al. use a 0-10 scale so perhaps say that your measure is very similar to Dohmen et al?)

Competitiveness measure: Is the competitive tournament performance compared to someone else's part 1 or part 2 performance? If the latter, anyone's or just those who chose to compete? I'm assuming that in cases of ties participants were paid?

Patience: Perhaps include a sentence on how this question has been validated, for those that do not know Falk et al? Are gender differences in patience robust? E.g. for risk preferences I know there is typically a gender difference but with patience I thought it was less clear.

Digit ratios: Just some more info here could be interesting – e.g. were there two raters per hand and what was there correlation? Or just one? Were there any “outliers” and how were these treated?

Figure 2: Is this the average of the two hands for 2D:4D?

Power discussion in the Discussion: In what units is 0.14? What about power and the other null results? Which effect sizes can be ruled out? What is the minimum detectable effect size for the various measures given the data? Or do you think these calculations do not make sense?

Referee: 2

Comments to the Author(s)

My review is included in the attached pdf file.

Author's Response to Decision Letter for (RSPB-2020-1756.R0)

See Appendix A.

RSPB-2020-1756.R1 (Revision)

Review form: Reviewer 1

Recommendation

Accept as is

Scientific importance: Is the manuscript an original and important contribution to its field?

Excellent

General interest: Is the paper of sufficient general interest?

Excellent

Quality of the paper: Is the overall quality of the paper suitable?

Excellent

Is the length of the paper justified?

Yes

Should the paper be seen by a specialist statistical reviewer?

No

Do you have any concerns about statistical analyses in this paper? If so, please specify them explicitly in your report.

No

It is a condition of publication that authors make their supporting data, code and materials available - either as supplementary material or hosted in an external repository. Please rate, if applicable, the supporting data on the following criteria.

Is it accessible?

Yes

Is it clear?

Yes

Is it adequate?

Yes

Do you have any ethical concerns with this paper?

No

Comments to the Author

The authors have done a great job addressing the comments that were raised. I would suggest that the p-values are kept in the main text for those of us who are interested in p-values.

Review form: Reviewer 2

Recommendation

Accept with minor revision (please list in comments)

Scientific importance: Is the manuscript an original and important contribution to its field?

Excellent

General interest: Is the paper of sufficient general interest?

Good

Quality of the paper: Is the overall quality of the paper suitable?

Good

Is the length of the paper justified?

Yes

Should the paper be seen by a specialist statistical reviewer?

No

Do you have any concerns about statistical analyses in this paper? If so, please specify them explicitly in your report.

No

It is a condition of publication that authors make their supporting data, code and materials available - either as supplementary material or hosted in an external repository. Please rate, if applicable, the supporting data on the following criteria.

Is it accessible?

No

Is it clear?

No

Is it adequate?

No

Do you have any ethical concerns with this paper?

No

Comments to the Author

See attached file.

Decision letter (RSPB-2020-1756.R1)

30-Oct-2020

Dear Dr van Leeuwen,

Your manuscript has now been peer reviewed and the reviews have been assessed by an Associate Editor. The reviewers' comments (not including confidential comments to the Editor) and the comments from the Associate Editor are included at the end of this email for your reference. As you will see, we are all agreed that the manuscript is close to acceptance, but one of the reviewers and the Associate Editor have made some suggestions that would greatly improve the manuscript and we would like to invite you to revise your manuscript to address them.

Research ethics:

Use of animals and field studies:

It is usually a condition of publication that you make available the data and research materials supporting the results in the article (<https://royalsociety.org/journals/authors/author-guidelines/#data>). However, we appreciate the circumstances of the confidentiality of the data, and are happy to accept the explanation for the lack of archived data.

Please submit a copy of your revised paper within three weeks. If we do not hear from you within this time your manuscript will be rejected. If you are unable to meet this deadline please let us know as soon as possible, as we may be able to grant a short extension.

Best wishes,
 Professor Loeske Kruuk
 Editor, Proceedings B
 mailto: proceedingsb@royalsociety.org

Associate Editor
 Comments to Author:

The manuscript has been re-reviewed by the original two experts in the field. Both reviewers found the manuscript to be much improved, and I agree with them. While Reviewer 1 judged that the manuscript would now be suitable for publication, Reviewer 2 still had a number of further comments. While we do not typically offer the opportunity for multiple rounds of review, I think that these further reviewer comments could usefully be addressed. In addition to the comments of both reviewers, I would add that I still felt that the structure of the manuscript could be a lot clearer. While the authors do now lay out two main aims at the end of the Introduction, it is hard to map these through the manuscript. The two aims are not used to structure either the methods or the results with subsections and subheadings. The Results section has seven subsections. To take the first three subsection titles, if analyzing "Testosterone at birth and adult 2D:4D", "Sex differences in economic preferences", and "Testosterone at birth and economic preferences" are all aims of the study, why not say so at the end of the introduction? A well-structured manuscript has a clear structure of aims/objectives or hypotheses with predictions that can be tracked easily through the manuscript. Analyses relating to one study aim should not be split across multiple results subsections as in the present manuscript - it just makes the study unnecessarily hard to follow.

Reviewer(s)' Comments to Author:
 Referee: 1

Comments to the Author(s)

The authors have done a great job addressing the comments that were raised. I would suggest that the p-values are kept in the main text for those of us who are interested in p-values.

Referee: 2

Comments to the Author(s)
 See attached file.

Author's Response to Decision Letter for (RSPB-2020-1756.R1)

See Appendix B.

Decision letter (RSPB-2020-1756.R2)

27-Nov-2020

Dear Dr van Leeuwen

I am pleased to inform you that your manuscript entitled "Do sex hormones at birth predict later-life economic preferences? Evidence from a pregnancy birth cohort study" has been accepted for publication in Proceedings B.

Open Access

You are invited to opt for Open Access, making your freely available to all as soon as it is ready for publication under a CCBY licence. Our article processing charge for Open Access is £1700. Corresponding authors from member institutions (<http://royalsocietypublishing.org/site/librarians/allmembers.xhtml>) receive a 25% discount to these charges. For more information please visit <http://royalsocietypublishing.org/open-access>.

Paper charges

Sincerely,
Professor Loeske Kruuk
Editor, Proceedings B

Associate Editor:

Board Member

Comments to Author:

(There are no comments.)

Appendix A

SCHOOL OF ECONOMICS AND MANAGEMENT

Dear Professor Kruuk,

Thank you for giving us the possibility to revise our manuscript (RSPB-2020-1756). We greatly appreciate the effort that you, the associate editor and the two referees have put into it. The questions, comments and recommendations that the associate editor and the referees raised helped us to improve the paper.

We have prepared a response letter with point-by-point replies to all the comments and suggestions raised by the associate editor and the referees. We believe that we have managed to follow the recommendations of the associate editor and the referees throughout.

Date
September 29, 2020

The main changes are:

Subject
Resubmission RSPB-2020-1756

Telephone
+31 13 466 21 47

E-mail
b.vanleeuwen@uvt.nl

1. We added two new subsections to the results section of the paper, where we discuss power, equivalence tests and multiple hypotheses testing.
2. We make clear in the methods and results sections of the manuscript how we followed the pre-registration, where we deviated and why.
3. We are more nuanced about our claims about the absence of observed associations. In particular, we propose to change the title to: "*No evidence for a link between sex hormones at birth and later-life economic preferences*".
4. We moved details on how we measured economic preferences to the supplementary materials in order to comply with the page limit.

Thank you again for your suggestions and the possibility to revise our manuscript. We are looking forward to hearing from you.

Yours sincerely,

Boris van Leeuwen
Paul Smeets
Jeanne Bovet
Gideon Nave
Jonathan Stieglitz
Andrew Whitehouse

Department of Economics

P.O. Box 90153 • 5000 LE Tilburg • The Netherlands • Visiting address • Warandelaan 2 • Tilburg • Telephone +31 13 466 91 11

IBAN NL74ABNA0602142539 • BIC ABNANL2A • VAT NL002791250B01

www.tilburguniversity.edu

Response to the Associate Editor

The manuscript has been reviewed by two experts in the field. Both reviewers like the topic and potential fit for the Journal. I am also generally supportive of the publication of large ‘failed-replication’ style manuscripts.

Thank you. We appreciate your time devoted to our manuscript and your constructive comments. They are very helpful in further strengthening our manuscript. We address all your comments point-by-point.

While Reviewer 1 has only minor suggestions for improvement, Reviewer 2 was quite critical, and thinks that extensive new analyses would be necessary before one could conclude that there were no effects in the data. As the analyses that are currently undertaken are quite simple, I have some sympathy for this position. At the same time, I do understand that the authors might be resistant to this, given that they had a pre-registration, including a pre-analysis plan. I think that a solution might be for the authors to undertake an additional set of analyses and present them in a different section. It would need to be made extremely clear in the MS that these additional analyses were undertaken to satisfy reviewer concerns, and were not part of the pre-registration. It is also more generally important for the authors to make it extremely clear how closely the pre-registration was followed (see Reviewer 1’s comments on this), and to flag any deviation from that pre-registration, no matter how small.

We agree with your proposed solution to present the additional analyses requested by Reviewer 2 in a new section. We now added subsections entitled “*Power analyses and equivalence tests*” and “*Multiple hypotheses testing corrections*” at the end of the results section (see pages 11-12) and made clear that these analyses were not part of the pre-registration. The main take away from these additional analyses is that they further corroborate our conclusion that we find no robust evidence for a link between sex hormones around birth and economic preferences.

We now also make clear in the methods section of the manuscript (see pages 7-8) how we followed the pre-registration, where we deviated and why. In particular, we now write:

“We pre-registered the main analyses and the way our variables are constructed. An important advantage of pre-registering for our study is that it restricts the degrees of freedom of researchers and precludes undisclosed flexibility in data analysis. The pre-analysis plan and the full experimental and survey design can be found at

https://osf.io/xt8s6/?view_only=eb37d6b404e94fd3b9c8952424d588f3

In the pre-analysis plan, we pre-registered 4 x 2 studies. We planned to study the relationship between neonatal hormones, stress and four types of outcome measures: (1) risk preferences, (2) social preferences, (3) competitiveness and (4) financial decision-making. In this paper, we report on the studies relating to neonatal hormones and risk preferences, social preferences and competitiveness. We added patience (from the financial decision-making part), as other papers have considered patience along with the other preferences as well.

We pre-registered four hormonal measures as the main independent variables of interest: total testosterone, BioT, the ratio between total testosterone and total estradiol (AE ratio), and the ratio between BioT and BioE2 (BioAE ratio). Hence, our pre-registered analyses are the 4 x 8 associations between the four hormone measures (total testosterone, BioT, AE ratio, BioAE ratio) and eight preference measures (incentivized risk tolerance, self-reported risk tolerance, competitiveness, patience, trust, positive reciprocity, negative reciprocity, dishonesty). In addition to these pre-registered analyses, we also report the associations between 2D:4D and preferences.

*In the pre-registration we stated that the main analyses would be ordinary least squares (OLS) regressions of the respective preferences measure on the respective hormone measure, a sex dummy and an interaction term between the hormone measure and the sex dummy. These analyses are reported in the SM. For expositional purposes, we deviate from the pre-analysis plan in the main text by reporting analyses without the sex*hormone interaction term, as well as associations between hormones and preferences by sex. Conclusions from results reported in the main text and SM are identical.”*

In addition to the comments of both reviewers, I would add that I felt that the structure of the manuscript could be clearer. I would cut the last paragraph of the Introduction, which unnecessarily summarizes the findings – save that for the Discussion.

We followed your suggestion and deleted the last paragraph of the introduction.

There could usefully be a clearer structure (e.g. aims or objectives, or hypotheses with predictions) laid out at the end of the Introduction that could be used to give the manuscript some explicit structure as subheadings in the Results section.

Thank you for this suggestion. The last paragraph of the introduction (page 4) now states the objectives of our study. Specifically, we added:

“Our study has two main objectives. First, we link direct measures of sex hormones at birth to later-life economic preferences ($200 \leq n \leq 217$, depending on the measure). Second, we test the robustness of earlier findings regarding the relationship between 2D:4D and economic preferences, but with a larger sample size ($533 \leq n \leq 597$, depending on the measure) than most previous studies. We compare our observed effects of sex hormones at birth and 2D:4D to sex differences in economic preferences in the same sample.”

We structure the results section according to these objectives.

Finally, I would add that details are missing on how individuals were characterized as men or women – by self-report? Please also be consistent in your terminology, I think you should use men and women throughout, not e.g. “females”.

Individuals are categorized by the sex that was determined at birth. We clarified this on page 6, by adding: *“We characterize study participants as men or women based on the biological sex that was determined at birth.”*

We now also use “men” and “women” throughout the text.

Response to Referee 1

This is a very interesting paper, resulting from a rigorously executed study on an important topic. I think it's particularly important to publish null results that come from high powered well designed studies like this one, especially when the literature is full of small studies with some "researcher degrees of freedom". The data on hormones at birth used here is really unique and any results are thus interesting. I recommend Accept and just have some minor comments. Addressing these could hopefully make the already very good paper even clearer. My comments appear below in their order of appearance.

Thank you for your fair assessment of our paper and your constructive comments, which helped to further strengthen the manuscript. We address all your comments point-by-point.

Abstract: I think it would be great to know something about the sample size or power already here.

In the revised manuscript, we included the sample sizes in the abstract. As the sample sizes vary somewhat between analyses, we opted to include the smallest numbers (n=200 for testosterone at birth, n=533 for 2D:4D). We discuss power in the new subsection "*Power analyses and equivalence tests*" (page 11) and mention power and equivalence tests in the abstract as well, where we write: "*Our sample size allows detecting effects larger than 0.22 SD for testosterone at birth, and 0.14 SD for 2D:4D ($\alpha=0.05$ and power=0.90). Equivalence tests show that for most associations, effects are unlikely to be larger than these bounds.*"

Pre-analysis plan: Is the pre-analysis plan fully followed? It would be informative if the main text could include a clear statement on this. In the SM it says "We pre-registered the main analysis and the way our variables are constructed." Can this be clarified so that the reader does not need to look at the pre-analysis plan but can understand what tests were pre-specified?

Thank you for this suggestion. We now discuss how we implemented the pre-analysis plan on pages 7-8 and when discussing the results, we are now more explicit about which analyses were not pre-registered. In particular, on pages 7-8 we now write:

"We pre-registered the main analyses and the way our variables are constructed. An important advantage of pre-registering for our study is that it restricts the degrees of freedom of researchers

and precludes undisclosed flexibility in data analysis. The pre-analysis plan and the full experimental and survey design can be found at

https://osf.io/xt8s6/?view_only=eb37d6b404e94fd3b9c8952424d588f3

In the pre-analysis plan, we pre-registered 4 x 2 studies. We planned to study the relationship between neonatal hormones, stress and four types of outcome measures: (1) risk preferences, (2) social preferences, (3) competitiveness and (4) financial decision-making. In this paper, we report on the studies relating to neonatal hormones and risk preferences, social preferences and competitiveness. We added patience (from the financial decision-making part), as other papers have considered patience along with the other preferences as well.

We pre-registered four hormonal measures as the main independent variables of interest: total testosterone, BioT, the ratio between total testosterone and total estradiol (AE ratio), and the ratio between BioT and BioE2 (BioAE ratio). Hence, our pre-registered analyses are the 4 x 8 associations between the four hormone measures (total testosterone, BioT, AE ratio, BioAE ratio) and eight preference measures (incentivized risk tolerance, self-reported risk tolerance, competitiveness, patience, trust, positive reciprocity, negative reciprocity, dishonesty). In addition to these pre-registered analyses, we also report the associations between 2D:4D and preferences.

*In the pre-registration we stated that the main analyses would be ordinary least squares (OLS) regressions of the respective preferences measure on the respective hormone measure, a sex dummy and an interaction term between the hormone measure and the sex dummy. These analyses are reported in the SM. For expositional purposes, we deviate from the pre-analysis plan in the main text by reporting analyses without the sex*hormone interaction term, as well as associations between hormones and preferences by sex. Conclusions from results reported in the main text and SM are identical.”*

Intro: Perhaps the authors could elaborate somewhat more on why these preferences are interesting to explore, besides in economic models?

On page 3 of the introduction we now explain how economic preferences are important besides economic models. We now write:

“In this paper we focus on economic preferences. Economic preferences – such as risk preferences, competitiveness, patience and social preferences – are the building blocks of economic models. While we label these preferences as economic preferences, they are clearly relevant for other domains of human decision-making. Economic preferences are related to many decisions in daily life (Sutter et al.

2013; Falk et al. 2018). Risk preferences influence health behavior, leisure activities and financial decisions (Dohmen et al. 2011). Competitiveness is associated with study choices and career paths (Buser et al. 2014). Social preferences predict whether people sacrifice resources to contribute to social welfare (Andreoni 1990; Charness and Rabin 2002) and predict socially responsible investment behavior (Riedl and Smeets 2017).”

Sample sizes: Some sample sizes are mentioned in the intro and some in the methods.

Perhaps explain the discrepancies?

In all analyses, we included all available data. As we have missing data for some of the variables (T, 2D:4D and economic preferences), sample sizes differ sometimes. We mention this in the methods section on page 8:

“In the regressions, we include all available data. This means that sample sizes may vary between regressions, because of missing data on some variables.”

Moreover, in the introduction (page 4) we now write:

“First, we link direct measures of sex hormones at birth to later-life economic preferences ($200 \leq n \leq 217$, depending on the measure). Second, we test the robustness of earlier findings regarding the relationship between 2D:4D and economic preferences, but with a larger sample size ($533 \leq n \leq 597$, depending on the measure) than most previous studies.”

Risk measure: Maybe I misunderstood something, but how are participants with multiple switching points treated?

As we use the Eckel-Grossman method to elicit risk attitudes in an incentivized way, multiple switching points are not possible. Participants selected one out of six lotteries to be played out, from which we infer their risk tolerance.

(Minor point – Dohmen et al. use a 0-10 scale so perhaps say that your measure is very similar to Dohmen et al?)

Thank you for spotting this. We now indicate that we use a different response scale than the original question used in Dohmen et al. To meet the page limit, we now discuss the details of the elicitation methods in the supplementary materials (SM). In the SM, on page 1, we now write:

“The self-reported measure is based on the validated question by Dohmen et al. (2011), which asks how willing people are to take risks in general. Participants answer using a 7-point Likert scale ranging from “completely unwilling to take risks” to “very willing to take risks” (whereas Dohmen et al. (2011) use a scale from 0 to 10).”

Competitiveness measure: Is the competitive tournament performance compared to someone else’s part 1 or part 2 performance? If the latter, anyone’s or just those who chose to compete? I’m assuming that in cases of ties participants were paid?

Performance is compared to someone else’s performance in part 2. This was a randomly selected other, so not necessarily someone who also chose to compete. This is different from the original Niederle-Vesterlund task, where there are 3 parts. Comparing the performance against a randomly selected other basically makes the choice of payment scheme an individual decision-making task, as the choice of the payment scheme does not affect others’ payments, just like in the original Niederle-Vesterlund task. As the experiment was conducted online, we opted for this simplified version of the Niederle-Vesterlund task. We based this on the experimental design of Buser, Geijtenbeek and Plug (2018). On page 3 of the SM, we now mention explicitly:

“If they choose ‘tournament’, their Part 2 score will be compared with that of a randomly selected other participant (who could have chosen either payment scheme).”

In case of a tie, participants were indeed paid (10 AUD per correctly solved puzzle), thank you for reading carefully and pointing this out. We now mention this on page 3 of the SM:

“... , in case of a tie they receive 10 AUD (7 US\$) per correctly solved puzzle.”

Patience: Perhaps include a sentence on how this question has been validated, for those that do not know Falk et al? Are gender differences in patience robust? E.g. for risk preferences I know there is typically a gender difference but with patience I thought it was less clear.

On page 3 of the SM we now say:

“The question has been validated by showing a positive correlation with incentivized time preference experiments, where individuals choose between a smaller amount that they receive earlier and a larger amount that they receive later (Falk et al. (2018)).”

You are right to note that the sex differences in patience are not clear-cut. We now provide a more nuanced discussion when discussing sex differences in time preferences. In the SM, when discussing sex differences, we now state on page 14 of the SM:

“For time preferences, the observation that men are more patient than women is consistent with previous findings by Falk et al. (2018), who use the same measure of time preferences as we do. Compared to the literature on sex differences and risk attitudes or competitiveness, there are much fewer studies on sex differences in time preferences and hence, the evidence is less clear-cut. For example, Sutter et al. (2013) and Almås et al. (2015) find no clear sex difference in time preferences, and Meier and Sprenger (2013) find that women are more patient than men.”

Digit ratios: Just some more info here could be interesting – e.g. were there two raters per hand and what was there correlation? Or just one? Were there any “outliers” and how were these treated?

In the dataset that we have access to, we only have measures from one rater. We included all data in the analyses, so also any potential outlier.

Figure 2: Is this the average of the two hands for 2D:4D?

In Figure 2, we show the estimates for both the left and the right hand 2D:4D separately (each in different shading). We added colors to the figure make the figure easier to read. Average 2D:4D is reported in Figure S5. In all cases, we find no significant association between 2D:4D and preferences.

Power discussion in the Discussion: In what units is 0.14? What about power and the other null results? Which effect sizes can be ruled out? What is the minimum detectable effect size for the various measures given the data? Or do you think these calculations do not make sense?

Thank you for stimulating us to think more about these issues. We now report more details on power analyses and what effect sizes can be ruled out in the new subsection *“Power analyses and equivalence tests”* (page 11). The reported effect sizes are in standard deviation terms (i.e. the effect, measured in SDs, of a 1 SD deviation increase in testosterone or 2D:4D). In the new subsection we now state:

“Thus far, our results generally indicate an absence of statistically significant associations between sex hormones at birth, 2D:4D and preferences. Following the suggestion of the reviewers, and beyond the analyses that we pre-registered, we further investigate what effect sizes are implausible based on our data. To this end, we conduct equivalence tests. Following Lakens (2017), we test whether our observed effect sizes are smaller than the smallest effect size we have sufficient power to detect. For testosterone at birth and risk tolerance, patience, reciprocity and dishonesty, our sample size of $n \geq 200$ allows us to detect effects larger than 0.22 SD units ($n = 200$, $\alpha = 0.05$ and power = 0.90), where this effect size is measured as the effect of a 1 SD increase in testosterone at birth on the respective preference. For binary preference measures (competitiveness and trust), we have sufficient power to detect effects larger than 0.11 percentage points. For 2D:4D, the minimal detectable effect size ($n = 533$, $\alpha = 0.05$ and power = 0.90) is equal to 0.14 SD units (risk tolerance, patience, reciprocity and dishonesty) or 0.07 percentage points (competitiveness and trust).

To test for equivalence, we use the “two one-sided tests” (TOST) procedure (see for example Lakens (2017)). Fig. S10 and Table S5 show the results. For the association between testosterone at birth and risk attitudes (both measures), trust, positive and negative reciprocity and dishonesty, we can reject effect sizes larger than the minimum detectable effect size. For the association between testosterone at birth and competitiveness and patience, we cannot reject that the effect is larger than the minimum detectable effect size. For 2D:4D and preferences, we can reject effect sizes larger than the minimum detectable effect size for all preference measures.”

Response to Referee 2

This study examined associations between sex hormones measured in umbilical cord blood at birth and economic preferences measured in an online study at ages 25 to 27, in approximately 200 participants. The study also reports analyses with digit ratios measured at ages 19 to 22, in approximately 500 participants. There are several strengths of the study, including the direct measurement of sex hormones in umbilical cord at birth with mass-spectrometry based methods, the longitudinal design, and the important research questions. Despite these strengths, I have major concerns with the paper in its current form, especially with the analyses reported and the conclusions drawn based on those analyses.

Thank you for your fair assessment of our paper and your constructive comments, which helped to further strengthen the manuscript. We address all your comments point-by-point.

Major issues

The authors draw the conclusion that there is “no link between sex hormones at birth and later-life economic preferences.” However, the results do not provide strong support for this conclusion for several reasons.

1. The study is underpowered to detect the small effect sizes expected for associations between sex hormones measured at birth and economic preferences later in life. A sample size of 200 has about 80% power to detect effect sizes of $r = .20$ or greater, but effect sizes are expected to be much smaller than this based on previous studies. Thus, a highly plausible interpretation of the results is that they represent Type II error - that there are small associations between sex hormones at birth and later-life economic preferences, but this study did not have sufficient statistical power to detect them. Type II error and power for the analyses involving direct measures of sex hormones do not seem to be discussed in the paper. The authors could also consider discussing other types of errors, such as Type S and M errors (Gelman & Carlin, 2014). I still think the data are valuable even though the study is underpowered (like many studies in neuroscience and neuroendocrinology, Button et al. 2013, Schultheiss et al., 2019), but these issues must be discussed carefully so readers can interpret your results accurately.

Thank you for these excellent suggestions. We now discuss those in the Discussion section on page 13:

“An important avenue for future research is to investigate the relation between sex hormones and economic preferences in larger samples. Studies in neuroscience and endocrinology often rely on small samples reducing the power of the conducted tests (Button et al., 2014; Parslow et al., 2019). In our study, we have hormone measures from umbilical cord blood for at least 200 participants (depending on the preference measure), which gives us 90% power to detect standardized effect sizes of 0.22 with a p-value of 0.05. This means that we lack the power to detect effects smaller than 0.22. We use a relatively large sample of 2D:4D measures (n = 533), which gives us 90% power to detect standardized effect sizes of 0.14 with a p-value of 0.05. However, if the true effect size of 2D:4D is smaller than 0.14 in magnitude, we still lack power to identify those effects. This could result in Type II errors.”

In the revised manuscript, we also followed your suggestion to be more nuanced about our claims. In particular, we changed the title to:

“No evidence for a link between sex hormones at birth and later-life economic preferences”.

We also created a new subsection “*Power analyses and equivalence tests*” in which we provide additional analyses to shed more light on power and detectable effect sizes (see our response to your second comment).

- 2. The authors cannot use $p > .05$ alone to draw the strong conclusion that there is “no link between sex hormones at birth and later-life economic preferences”. This wording implies you are accepting the null hypothesis based on $p > .05$, which is a misinterpretation of what the statistics you report are telling you. As Lakens 2017 states: “It is statistically impossible to support the hypothesis that a true effect size is exactly zero.” The paper would benefit from additional analyses, such as Bayesian analyses, to determine the extent to which the results are more strongly in favor of the null or alternative hypothesis (Dienes, 2014). Equivalence tests may also be useful if you choose to stick to a frequentist statistics framework (Lakens, 2017).**

Thank you for these suggestions. Indeed, the absence of evidence for an effect is not evidence of its absence. We followed your suggestion to conduct equivalence tests, which we report in the new subsection “*Power analyses and equivalence tests*” (page 11). In the new subsection we now state:

“Thus far, our results generally indicate an absence of statistically significant associations between sex hormones at birth, 2D:4D and preferences. Following the suggestion of the reviewers, and beyond the analyses that we pre-registered, we further investigate what effect sizes are implausible based on our data. To this end, we conduct equivalence tests. Following Lakens (2017), we test whether our observed effect sizes are smaller than the smallest effect size we have sufficient power to detect. For testosterone at birth and risk tolerance, patience, reciprocity and dishonesty, our sample size of $n \geq 200$ allows us to detect effects larger than 0.22 SD units ($n = 200$, $\alpha = 0.05$ and power = 0.90), where this effect size is measured as the effect of a 1 SD increase in testosterone at birth on the respective preference. For binary preference measures (competitiveness and trust), we have sufficient power to detect effects larger than 0.11 percentage points. For 2D:4D, the minimal detectable effect size ($n = 533$, $\alpha = 0.05$ and power = 0.90) is equal to 0.14 SD units (risk tolerance, patience, reciprocity and dishonesty) or 0.07 percentage points (competitiveness and trust).

To test for equivalence, we use the “two one-sided tests” (TOST) procedure (see for example Lakens (2017)). Fig. S10 and Table S5 show the results. For the association between testosterone at birth and risk attitudes (both measures), trust, positive and negative reciprocity and dishonesty, we can reject effect sizes larger than the minimum detectable effect size. For the association between testosterone at birth and competitiveness and patience, we cannot reject that the effect is larger than the minimum detectable effect size. For 2D:4D and preferences, we can reject effect sizes larger than the minimum detectable effect size for all preference measures.”

Related to this issue of misinterpretation of results, the authors are also encouraged to think in terms of effect size estimation (point estimates and confidence intervals) rather than p less than or greater than .05 alone; the latter type of binary thinking can lead to erroneous conclusions (Cumming, 2014; Amrhein et al., 2019; McShane et al., 2019). Our best guess for a particular effect is the point estimate. From this perspective, your results suggest that there may be small effects (small effects are expected based on what we know about sex hormones at birth and outcomes later in life) and that additional studies with larger sample sizes will be required to increase the precision in the estimates you report. For the reasons stated here, phrases like we estimate “precise null effects”, “no evidence”, and “no link” are all inappropriate and are not strongly supported by the results. More nuanced conclusions focused on the direction, size, and uncertainty in estimates are more appropriate, including in prominent places like the abstract.

We agree, and followed your suggestion to be more nuanced about our claims across the manuscript, including the abstract and title, which we propose to change to: *“No evidence for a link between sex hormones at birth and later-life economic preferences”*. We also removed the somewhat ambiguous claim about “precise null effects” from the abstract and other places in the text.

We further agree with you that additional studies with larger sample sizes will be very useful to shed more light on the relation between sex hormones at birth and later-life economic preferences. We now clarify this in the discussion section on page 13:

“An important avenue for future research is to investigate the relation between sex hormones and economic preferences in larger samples. Studies in neuroscience and endocrinology often rely on small samples reducing the power of the conducted tests (Button et al., 2014; Parslow et al., 2019). In our study, we have hormone measures from umbilical cord blood for at least 200 participants (depending on the preference measure), which gives us 90% power to detect standardized effect sizes of 0.22 with a p-value of 0.05. This means that we lack the power to detect effects smaller than 0.22. We use a relatively large sample of 2D:4D measures (n = 533), which gives us 90% power to detect standardized effect sizes of 0.14 with a p-value of 0.05. However, if the true effect size of 2D:4D is smaller than 0.14 in magnitude, we still lack power to identify those effects. This could result in Type II errors.”

We also refer to this in the abstract, to which we added:

“Our sample size allows detecting effects larger than 0.22 SD for testosterone at birth, and 0.14 SD for 2D:4D ($\alpha=0.05$ and power=0.90). Equivalence tests show that for most associations, effects are unlikely to be larger than these bounds.” Our results suggest a reinterpretation of results from prior 2D:4D studies predicting economic preferences, and other outcomes. Our results also highlight the importance of future large sample studies that permit detection of small effects.”

Regarding your suggestion of confidence intervals, we already included these in the manuscript alongside p-values, both in the text and the figures. We favor the approach to keep both the CI and p-values, but are open to dropping the p-values.

- 3. There is ambiguity about what criteria the authors are using to draw conclusions about whether a particular effect is real/non-zero, and this leads to additional questions about the main conclusions. For example, the authors report two “significant” associations between sex hormones at birth and economic preferences later in life. One of these**

significant effects is reported in Figure 2, and the other is reported in Figure S4. At first glance, these “significant” effects seem at odds with the simplistic conclusion that there is “no link between sex hormones at birth and later-life economic preferences.” In the discussion, the authors talk about only one of these effects and conclude that “this is likely a false positive, given the number of tests performed.” The authors do not mention the other significant effect reported in Figure S4, and it not clear why. The authors are also ambiguous about the key number of tests that are being used to support this false positive conclusion. Based on the information the authors report, I am not fully convinced that the “significant” effects are definitely false positives. Several pieces of evidence suggest that these effects could be real and may be worthy exploring further in new studies. First, the significant association between total T at birth and competitiveness reported in Figure 2 replicates when using bioavailable T at birth as reported in Figure S4, when using $\ln(\text{total T at birth})$ as reported in Figure S6, and in an analysis of men only as reported in Figure 3. Second, the significant association between total AE ratio at birth and trust shows a very similar pattern and seems to just barely miss statistical significance when looking at bioavailable AE ratio, as reported in Figure S4.

Thank you for pointing this out. We now also discuss these additional significant findings on pages 9-10, where we write:

“Fig. S3 shows associations between other hormonal measures that we pre-registered (BioT, AE ratio and BioAE ratio) and preferences, again controlling for sex. These results are comparable to total testosterone at birth. We find no robust associations between any of these hormonal measures and preferences. Out of 24 associations, we only find a significant (negative) association between BioT and competitiveness ($b = -0.070$, $t(201) = -2.11$, 95% CI = $[-0.135, -0.005]$, $n = 204$, $p = 0.036$), and a significant (positive) association between the AE ratio and trust ($b = 0.065$, $t(205) = 1.98$, 95% CI = $[0.000, 0.130]$, $n = 208$, $p = 0.049$).”

In the discussion section, we no longer claim that the positive effects we found could be false positives, but rather included an additional discussion on sample size, power, Type S and Type M errors and point towards directions for future research. On page 13, we now write:

“An important avenue for future research is to investigate the relation between sex hormones and economic preferences in larger samples. Studies in neuroscience and endocrinology often rely on small samples reducing the power of the conducted tests (Button et al., 2014; Parslow et al., 2019). In our study, we have hormone measures from umbilical cord blood for at least 200 participants (depending on the preference measure), which gives us 90% power to detect standardized effect sizes

of 0.22 with a p-value of 0.05. This means that we lack the power to detect effects smaller than 0.22. We use a relatively large sample of 2D:4D measures ($n = 533$), which gives us 90% power to detect standardized effect sizes of 0.14 with a p-value of 0.05. However, if the true effect size of 2D:4D is smaller than 0.14 in magnitude, we still lack power to identify those effects. This could result in Type II errors. Small sample sizes cannot only result in Type I and Type II error, but also in errors of Type S (wrong sign) and Type M (overestimation of magnitude) (Gelman and Carlin, 2014). In our case, we sometimes find a negative association between testosterone and competitiveness, running the opposite direction than expected, and could result from Type S error. Studies with larger samples will have more power to make more definitive claims. It is also interesting to establish relevant moderators for the effects of hormones on outcome measures (see e.g. Schultheiss et al. (in press)). For example, one moderator could be levels of adult sex hormones. Moreover, umbilical cord blood measures of sex hormones are relatively new in the literature. We hope that future research will provide additional validation tests of this measure. These validation studies will provide a stronger basis for future studies using umbilical cord blood measures.”

It is also possible this AE ratio association with trust could be somewhat stronger in men, but I did not see where such an analysis is reported (apologies if I missed this).

We did not report the associations between the AE ratio and preferences by sex in the previous version. In the new version, we report these analyses (as well as for bioavailable T and the bioavailable AE ratio) in Figure S4 and Table S3, and mention them on page 10, where we write:

“Fig. S4 shows the associations for BioT, the AE ratio and the BioAE ratio by sex. For men, we find a significant (negative) association between BioT and competitiveness and a significant (positive) association between the AE ratio and trust. All other 46 associations, for either sex, are nonsignificant.”

Indeed, you are right that the significant associations between the AE ratio and trust and between bioavailable T and competitiveness appear to be driven by men: the associations for women are not statistically significant at the 5% level.

Third, the authors state that they do not formally correct for multiple comparisons in their analyses, so it is unclear why they later go on to informally correct for multiple comparisons in the discussion when interpreting their results. The reader is left confused as to what criteria the authors are using to evaluate whether a particular effect is a false

positive, and why these criteria are being used. What if the authors had found two significant effects in the middle panel of Figure 2? Would the second significant effect also have been considered a false positive? To address this ambiguity, it may be helpful to report formal corrections for multiple comparisons or at least explain to the reader in more detail the rationale for their false positive interpretation. After reading your paper, it may be reasonable for a reader to believe that the primary analyses that should go into multiple comparison corrections are the eight associations between total T and economic outcome measures reported in the middle panel of Figure 2, because the paper is written and presented as if these are the eight tests of primary interest. But it is possible that the authors believe a small or larger number of effects should go into the corrections for multiple comparisons. An alternative possibility is to switch to Bayesian multilevel models as a better solution to address the multiple comparisons issue, as recommended by some statisticians (Gelman, Hill, & Yajima, 2012).

We agree with you that it is helpful to formally correct for multiple hypotheses testing. In the pre-registration we indicated to focus on eight main preference measures and four main hormone measures (total T, total AE ratio, BioT, BioAE ratio).

In total this results in $8 \times 4 = 32$ tests. We did not pre-register multiple comparison correction, yet see three possible correction levels to be justifiable, based on 32 (alpha corrected for all possible combinations), 8 (alpha corrected for all possible preferences measure across hormonal measure) or 4 (alpha corrected for all possible hormonal measures across preferences measures) comparisons. To ensure that our conclusions would not be driven by the flexibility that this analytical choice allows, we report adjusted p-values (“q-values”) based on 32, 8 or 4 comparisons, both using Bonferroni corrections and False Discovery Rate (FDR) corrections (in Table S6). We find that only when we correct for four comparisons, the significant association between T and competitiveness remains significant at the $q < 0.05$ (corrected) level. For all other possibilities, the previously significant effects become nonsignificant. We discuss this in the new subsection “*Multiple hypotheses testing corrections*” on pages 11-12, where we write:

“Following our pre-analysis plan, we did not correct for multiple hypothesis testing. We pre-registered 32 tests: eight preference measures times four hormone measures. Out of these 32 tests, three resulted in a p-value smaller than 0.05. We did not pre-register multiple comparison correction, yet see three justifiable corrections for multiple hypotheses testing, based on 32 comparisons (correct for all tests), 8 comparisons (correct for all preferences measures across hormonal measures) or 4

comparisons (correct for all hormonal measures across preferences measures). In Table S6, we report adjusted p-values (“q-values”) based on 32, 8 or 4 comparisons, both using Bonferroni corrections and False Discovery Rate (FDR) corrections (Simes 1986; Benjamini and Hochberg 1995). The (negative) association between total testosterone at birth and competitiveness remains statistically significant if we correct for 4 comparisons but not if we correct for 8 or 32 comparisons. The (negative) association between BioT and competitiveness and the (positive) association between the AE ratio and trust are both no longer statistically significant, regardless of whether we correct for 4, 8 or 32 comparisons.”

Finally, please note that our multiple hypotheses corrections only focus on the main analyses. The 2D:4D analyses are not part of the main analyses that we specified in the pre-analysis plan. Moreover, as all associations between 2D:4D and preferences are nonsignificant without correcting for multiple comparisons, such corrections would not affect our results.

- 4. It seems strange that the authors have not also reported analyses for total estradiol and bioavailable estradiol. It seems they only consider estradiol in ratio analyses. Without these analyses, I do not think the authors can draw the strong conclusion that there is “no link between sex hormones at birth and later-life economic preferences”. Perhaps the primary conclusions should be adjusted to match the analyses that are reported, or exploratory analyses with total estradiol and bioavailable estradiol can be added to the paper to evaluate the main conclusion with greater care.**

Thank you for this suggestion. In the previous version of the paper, we focused on the hormone measures that we pre-registered as our main analyses. These were total T, bioavailable T, the total AE ratio and the bioavailable AE ratio. In the revised paper, we included additional analyses linking total estradiol and bioavailable estradiol to our eight preference measures. Figure S9 shows the results. None of the associations between total estradiol or bioavailable estradiol and economic preferences are statistically significant. We discuss this on page 11, where we write:

“Also, for estrogen measures (estradiol (E2) and BioE2), none of the associations between estrogens and preferences is statistically significant.”

- 5. Another interpretation that would go against your conclusion that there is “no link between sex hormones at birth and later-life economic preferences” is that there could be a moderator that you did not consider. For example, Schultheiss et al. (in press;**

<https://osf.io/xm8t6/>) provide some evidence for a moderated mediation effect: a sex difference in narrative-writing fluency is mediated by adult levels of estradiol, and this mediation effect is moderated by perinatal hormone exposure (measured with digit ratio, with the limitations of this measure acknowledged in the paper). This type of moderated mediation model could be tested in your study if you also had hormone measurements around the same time that you measured economic preferences. If you do not have these measures or consider them outside the scope of your paper, you could still consider the possibility of moderators like adult hormone levels in the discussion section of your paper and adjust conclusions in prominent places like the abstract. For example, your abstract could say that your discussion focuses on multiple explanations for the results.

Thank you for the suggestion to explore moderators. Like you mentioned, we would need hormonal measurements around the same time that we measured economic preferences. Unfortunately, we do not have such measures. We therefore followed your suggestion to include discussion of the importance of moderators. In the discussion on page 13, we included:

“It is also interesting to establish relevant moderators for the effects of hormones on outcome measures (see e.g. Schultheiss et al. (in press)). For example, one moderator could be levels of adult sex hormones.”

We choose for the discussion section to prevent making the abstract too long, but are open to include this in the abstract as well.

Additional suggestions

- 6. Sample size for the primary analyses is a crucial piece of information for interpreting the results that should be reported in the abstract.**

We included the sample sizes in the abstract now. As the sample sizes vary somewhat between analyses, we opted to include the smallest numbers (n=200 for testosterone at birth, n=533 for 2D:4D).

- 7. Because the umbilical cord measure of hormones is relatively understudied within this literature (a clear strength of your work), it could be beneficial to have a more detailed section on the validity of this measure perhaps in the supplemental material. For example, this section could explain what do we and do not know about the correlations between**

this measure and hormone levels from other fluids, including amniotic fluid, hormone levels in mom’s blood captured from a normal blood draw, child hormone levels after birth, adulthood hormone levels, cerebrospinal fluid hormone levels, etc. Any relevant studies with fetal tissue in non-human animals could also be relevant to this section. Any associations between this umbilical cord measure and markers of physical development or psychological outcomes in babies or young children could also be included. This section could also make recommendations for future validation work.

Umbilical cord blood is a relatively new measure of sex hormones. This means that the body of evidence is still relatively scant. We already included the relevant studies that we are aware of, which we discuss on page 4, where we write:

“Our study therefore uses direct measures of neonatal sex hormones collected at birth from umbilical cord blood. Umbilical cord blood is thought to reflect hormone exposure during late gestation (Hollier et al. 2014). Robust sex differences in cord blood hormone measures are found (Barry et al. 2011; Keelan et al. 2012). It is possible that critical periods for hormone exposure and neurodevelopment occur earlier in gestation. While causal evidence for humans is lacking, nonhuman animal models suggest that separate critical periods exist for genital and neurodevelopmental effects of hormone exposure. Importantly, late gestation was found to be a critical period for neurodevelopment (Goy et al. 1988; Roselli et al. 2011). Umbilical cord blood measures can thus provide important new insights.”

We now also provide suggestions for future validation studies in the discussion section. On page 13 it now reads:

“Moreover, umbilical cord blood measures of sex hormones are relatively new in the literature. We hope that future research will provide additional validation tests of this measure. These validation studies will provide a stronger basis for future studies using umbilical cord blood measures.”

8. The authors should openly acknowledge the limitations of their study and recommend future directions to address these limitations.

We followed your suggestions and included limitations and directions for future research both with regard to the validation of umbilical cord blood measures (see our response to your previous comment) and with regard to studies on sex hormones and economic preferences (see the response to comments 2 and 5). These points can be found in the discussion section on pages 12-13. In particular, we write:

“... Turning to our umbilical cord blood measure, it is possible that critical periods for hormone exposure and neurodevelopment occur early in gestation. While different measures of sex hormones around birth all have their limitations, including ours, it is crucial to find convergent evidence across measures. More than 1,400 studies rely on 2D:4D to assess fetal hormonal exposure, which makes it essential to test convergent validity of this measure against others.

An important avenue for future research is to investigate the relation between sex hormones and economic preferences in larger samples. Studies in neuroscience and endocrinology often rely on small samples reducing the power of the conducted tests (Button et al., 2014; Parslow et al., 2019). In our study, we have hormone measures from umbilical cord blood for at least 200 participants (depending on the preference measure), which gives us 90% power to detect standardized effect sizes of 0.22 with a p-value of 0.05. This means that we lack the power to detect effects smaller than 0.22. We use a relatively large sample of 2D:4D measures (n = 533), which gives us 90% power to detect standardized effect sizes of 0.14 with a p-value of 0.05. However, if the true effect size of 2D:4D is smaller than 0.14 in magnitude, we still lack power to identify those effects. This could result in Type II errors. Small sample sizes cannot only result in Type I and Type II error, but also in errors of Type S (wrong sign) and Type M (overestimation of magnitude) (Gelman and Carlin, 2014). In our case, we sometimes find a negative association between testosterone and competitiveness, running the opposite direction than expected, and could result from Type S error. Studies with larger samples will have more power to make more definitive claims. It is also interesting to establish relevant moderators for the effects of hormones on outcome measures (see e.g. Schultheiss et al. (in press)). For example, one moderator could be levels of adult sex hormones. Moreover, umbilical cord blood measures of sex hormones are relatively new in the literature. We hope that future research will provide additional validation tests of this measure. These validation studies will provide a stronger basis for future studies using umbilical cord blood measures.”

- 9. I appreciate the fact that the study was pre-registered, but pre-registration is not a panacea. Now that pre-registration is becoming more common, it is becoming clear that pre-registrations do not solve all problems, can serve many goals, and vary in their ability to constrain hypotheses, analyses, and results interpretation. Thus, rather than only saying the study was pre-registered and pointing to a link, the paper would benefit from the authors explaining what they think the benefits of the pre-registration were for this particular study relative to a situation in which they did not pre-register. This discussion**

could potentially be included in the supplemental material, with recommendations to improve pre-registrations in future studies.

You are right that pre-registration does not solve all problems. The main advantage of pre-registering for our study is, in our view, that it restricts the degrees of freedom of researchers and precludes undisclosed flexibility in data analysis. In pre-registering the study, we spent most effort on being explicit about the main analyses and the way our variables are constructed. In the revised manuscript, on pages 7-8, we discuss the pre-registration plan in more detail and write:

“We pre-registered the main analyses and the way our variables are constructed. An important advantage of pre-registering for our study is that it restricts the degrees of freedom of researchers and precludes undisclosed flexibility in data analysis. The pre-analysis plan and the full experimental and survey design can be found at

https://osf.io/xt8s6/?view_only=eb37d6b404e94fd3b9c8952424d588f3

In the pre-analysis plan, we pre-registered 4 x 2 studies. We planned to study the relationship between neonatal hormones, stress and four types of outcome measures: (1) risk preferences, (2) social preferences, (3) competitiveness and (4) financial decision-making. In this paper, we report on the studies relating to neonatal hormones and risk preferences, social preferences and competitiveness. We added patience (from the financial decision-making part), as other papers have considered patience along with the other preferences as well.

We pre-registered four hormonal measures as the main independent variables of interest: total testosterone, BioT, the ratio between total testosterone and total estradiol (AE ratio), and the ratio between BioT and BioE2 (BioAE ratio). Hence, our pre-registered analyses are the 4 x 8 associations between the four hormone measures (total testosterone, BioT, AE ratio, BioAE ratio) and eight preference measures (incentivized risk tolerance, self-reported risk tolerance, competitiveness, patience, trust, positive reciprocity, negative reciprocity, dishonesty). In addition to these pre-registered analyses, we also report the associations between 2D:4D and preferences.

*In the pre-registration we stated that the main analyses would be ordinary least squares (OLS) regressions of the respective preferences measure on the respective hormone measure, a sex dummy and an interaction term between the hormone measure and the sex dummy. These analyses are reported in the SM. For expositional purposes, we deviate from the pre-analysis plan in the main text by reporting analyses without the sex*hormone interaction term, as well as associations between hormones and preferences by sex. Conclusions from results reported in the main text and SM are identical.*

In the regressions, we include all available data. This means that sample sizes may vary between regressions, because of missing data on some variables. All p-values reported in the text are based on OLS regressions (unless stated otherwise) and are not corrected for multiple hypotheses testing.”

SCHOOL OF ECONOMICS AND MANAGEMENT

Dear Professor Kruuk,

Thank you for giving us the opportunity to revise and resubmit our manuscript (RSPB-2020-1756.R1). We greatly appreciate the time and effort that you, the associate editor and the two referees have put into it. We used your comments effectively to revise the paper.

Attached you will find a response letter with point-by-point replies to all the comments and suggestions raised by the associate editor and the referees. We believe that we have managed to follow the recommendations of the associate editor and the referees throughout.

Date

November 17, 2020

Subject

Resubmission RSPB-2020-1756.R1

Telephone

+31 13 466 21 47

E-mail

b.vanleeuwen@uvt.nl

The main changes are:

1. We reorganized the Results section to follow our main study aims more closely. We think this has improved the flow of the manuscript, and thank the Associate Editor for this suggestion.
2. We substantially revised the Discussion section. We now discuss what effect sizes could have been expected based on the literature and we included a “constraints on generality” statement.
3. Throughout the manuscript, we are more precise about our main findings, avoiding claims of “no evidence” and rather discussing the range of estimated effect sizes. In particular, we propose to change the title to: *“Do sex hormones at birth predict later-life economic preferences? Evidence from a pregnancy birth cohort study”*.

Thank you again for your suggestions and the opportunity to revise and resubmit our manuscript. We believe the manuscript has been substantially improved compared to the previous version. We are looking forward to hearing from you.

Yours sincerely,

Boris van Leeuwen
Paul Smeets
Jeanne Bovet
Gideon Nave
Jonathan Stieglitz
Andrew Whitehouse

Department of Economics

P.O. Box 90153 • 5000 LE Tilburg • The Netherlands • Visiting address • Warandelaan 2 • Tilburg • Telephone +31 13 466 91 11

IBAN NL74ABNA0602142539 • BIC ABNANL2A • VAT NL002791250B01

www.tilburguniversity.edu

Response to the Associate Editor

Comments to Author:

The manuscript has been re-reviewed by the original two experts in the field. Both reviewers found the manuscript to be much improved, and I agree with them. While Reviewer 1 judged that the manuscript would now be suitable for publication, Reviewer 2 still had a number of further comments. While we do not typically offer the opportunity for multiple rounds of review, I think that these further reviewer comments could usefully be addressed. In addition to the comments of both reviewers, I would add that I still felt that the structure of the manuscript could be a lot clearer. While the authors do now lay out two main aims at the end of the Introduction, it is hard to map these through the manuscript. The two aims are not used to structure either the methods or the results with subsections and subheadings. The Results section has seven subsections. To take the first three subsection titles, if analyzing "Testosterone at birth and adult 2D:4D", "Sex differences in economic preferences", and "Testosterone at birth and economic preferences" are all aims of the study, why not say so at the end of the introduction? A well-structured manuscript has a clear structure of aims/objectives or hypotheses with predictions that can be tracked easily through the manuscript. Analyses relating to one study aim should not be split across multiple results subsections as in the present manuscript - it just makes the study unnecessarily hard to follow.

Our response:

Thank you for your constructive comments. We are glad that you find the manuscript to be much improved.

Following your suggestion, we reorganized the Results section. We begin the Results section with a subsection on descriptive analyses, highlighting sex differences in hormones and in preferences (we moved some analyses (Fig. 1 in the previous round) within this subsection to the SM to keep the focus on the main aims). We then present the two main aims ("*Testosterone at birth and economic preferences*" and "*2D:4D and economic preferences*"). We conclude the Results section with the additional analyses suggested by the referees in the previous round. We think this has improved the flow of the manuscript, and thank you for this suggestion.

Response to Referee 1

Comments to the Author(s)

The authors have done a great job addressing the comments that were raised. I would suggest that the p-values are kept in the main text for those of us who are interested in p-values.

Our response:

Thank you again for your constructive comments in the previous rounds. We are glad that you feel that the paper is ready to be published, and addressed your comment above by keeping the p-values in the main text.

Response to Referee 2

Comments to the Author(s)

The authors did a good job of addressing many of my comments. I feel that the paper is substantially improved. I also reiterate that this study has several strengths, especially its use of measurement of sex hormones in umbilical cord blood. Despite the many strengths of the revised paper, I still have some suggestions for improving the paper further. Many of the comments revolve around how to interpret the results of an underpowered study.

Our response:

Thank you again for your constructive comments. We are glad that you feel the paper has substantially improved. Below, we respond to your remaining suggestions.

Primary concern

1. In the new discussion paragraph, the authors say “This means we lack the power to detect effects smaller than 0.22.” Unfortunately, the paper is missing any serious discussion about various effect sizes one might expect based on previous research. I did not see anything about this in the methods either, or in the pre-registration (e.g. a power analysis or a power curve). Apologies if I missed this. Why should you have expected effect sizes of 0.22 or greater? A quick skim of some papers in this literature suggests smaller effect sizes are expected, and therefore, that this study is underpowered. If so, then as I said in my prior review, sticking to the null hypothesis significance testing (NHST) framework alone to draw conclusions like “no evidence” is very misleading because a low-powered study by definition will not be able to detect $p < .05$ very often. An estimation framework is much better for drawing appropriate conclusions. A serious transparent discussion about this issue would improve your paper a lot and make sure readers do not make mistakes when interpreting your results.

Our response:

Thank you for these suggestions. As our study is the first to relate sex hormones from umbilical cord blood to economic preferences, there is no previous literature to directly compare our data to. Closest to our work is the literature on 2D:4D and economic preferences. We now include a discussion of effect sizes in this literature. On pages 12-13 we now mention:

“In our study, we have hormone measures from umbilical cord blood for over 200 participants (depending on the preference measure), which gives us 90% power to detect effect sizes of 0.22 SD ($\alpha = 0.05$, two sided). This means that we lack the power to detect effects smaller than 0.22 SD. We use a relatively large sample of 2D:4D measures ($n = 533$), which gives us 90% power to detect effect sizes of 0.14 SD ($\alpha = 0.05$, two sided). Based on previous papers, it is plausible that effect sizes are smaller than this. For example, for risk attitudes, Brañas-Garza et al. (2018) report an effect of 0.08 SD ($n=543$, based on our calculations of results reported in Brañas-Garza et al. (2014)), Bönnte et al. (2016) report an effect size of 0.10 SD ($n=432$) and Garbarino et al. (2011) report an effect of 0.16 SD ($n=151$, based on our calculations). To reliably detect such small effects, larger samples are needed. For example, to detect an effect of 0.08 SD with 90% power ($\alpha = 0.05$) a sample size of $n=1,600$ is required. This means that

previously detected small effects come from low-powered studies, which could lead to overestimation of effect sizes (type-M errors, Gelman and Carlin 2014). It is important to note that many studies also report null effects. Parslow et al. (2019) survey the literature on 2D:4D and risk attitudes and find that based on 18 papers, 93 out of 109 reported tests are nonsignificant. Studies with larger samples will have more power to make more definitive claims.”

Regarding ex-ante power analysis: we did not conduct any ex-ante power calculations, because we invited all people in the Raine Study to participate in our experiment. Thus, our goal was to maximize our power by recruiting as many participants with neonatal hormone measures as possible, rather than aiming for a specific sample size. We now mention this on page 8:

“We did not perform any ex-ante power calculations, as we aimed to maximize our power by inviting all active participants in the Raine Study.”

Other points

2. The title is better than the previous one, but I disagree that the results indicate “no evidence for a link...”. There is some evidence (e.g. the equivalence tests showed some ambiguity, there were some significant results, the point estimates are not all close to 0.000000), but the evidence is weak and ambiguous. This is not surprising because the study is underpowered to detect a range of plausible effect sizes. The point estimates you found make sense. Based on a skim of some prior papers, we would not expect large effect sizes for this type of study. The patience results shown in Fig. 2 are a good example of how null hypothesis significance testing alone can lead to misleading conclusions like “no evidence for a link”. You found a “significant” sex difference in patience, but a “non-significant” association between sex hormones and patience even though the point estimates were very similar and the direction and size of these point estimates make sense. The non-significant association here may be explained by the lower power in the sex hormone analysis compared to the sex difference analysis. That is, there could very well be an association between sex hormones at birth and patience in line with the point estimate you found. Therefore, to avoid drawing strong erroneous conclusions, I would suggest a different more accurate title that focuses on estimation rather than statistical significance, e.g. “Estimating associations between sex hormones at birth and later-life economic preferences”. And throughout the paper, you could avoid strong claims about no evidence and replace with or add some statements related to effect size estimation (point estimates and confidence intervals).

Our response:

Following your suggestion, we removed all claims of “no evidence” and discuss the range of estimated effect sizes instead. We propose to change the title to “*Do sex hormones at birth predict later-life economic preferences? Evidence from a pregnancy birth cohort study*”. This new title should be more neutral regarding the paper’s findings.

3. Abstract: It would be helpful to add information about the effect sizes you found in the abstract (e.g. point estimates generally ranging from X to Y), especially for the association between umbilical cord measures of sex hormones and economic preference.

Our response:

We agree with you that this information is important to communicate. To comply with the word limit (200 words) we substantially revised the abstract. We now mention in the abstract:

“We find no significant associations between testosterone at birth and preferences, except for competitiveness, where the effect runs opposite to the expected direction. Point estimates are between 0.05-0.09 percentage points (pp) and 0.003-0.14 SD. We similarly find no significant associations between 2D:4D and preferences (n=533, point estimates 0.003-0.02 pp and 0.001-0.06 SD).”

At the beginning of the discussion section, we now also discuss the range of estimated effect sizes in more detail (see our response to your next comment).

4. The same point applies to the beginning of the discussion section. It would be helpful to talk about the range of effect sizes you found, which is more helpful than saying “no robust evidence” and will move the paper more towards an estimation framework. With an underpowered study, it is by definition unlikely that would have found robust evidence.

Our response:

We changed the beginning of the Discussion section. We no longer mention “no robust evidence” and mention our main findings, including the range of estimated effect sizes instead. It now reads:

“We estimate associations between neonatal sex hormones and later-life economic preferences. Unlike previous research, we use direct measures of neonatal sex hormone concentrations, measured from umbilical cord blood collected at birth. We find no significant associations between total testosterone at birth and preferences, except for competitiveness. However, this effect runs opposite to the expected direction. The point estimates are between 0.05 and 0.09 percentage points for binary preference measures (competitiveness and trust) and between 0.001 and 0.14 SD for non-binary preference measures (risk tolerance, patience, positive/negative reciprocity, dishonesty). We also examine the widely used 2D:4D digit ratio in a sample larger than most previous studies. For 2D:4D, we estimate null effects with point estimates between 0.003 and 0.02 percentage points and 0.001 and 0.06 SD.”

5. Introduction: “when the children were between 25-27 years old” is awkward wording because 25-27 years old is considered adulthood.

Our response:

Thank you for spotting this. We now write on page 4:

“In the present study we conducted incentivized experiments measuring economic preferences of the offspring (Raine Study Gen2) when they were 25-27 years old.”

6. Prior to publication, if the authors consider it feasible and appropriate, I strongly suggest that they make their dataset and analysis stream open (e.g. on the open science framework) and include links in the paper. I saw links to the pre-registration, but not to an open dataset or full transparent analysis stream. But I may have missed this. This would be in line with the push toward open science and has many advantages relative to a closed dataset and analysis stream. If the authors do not think this is feasible or appropriate, they could potentially include a brief explanation as to why they are not doing this in the paper and also add text to the paper encouraging future studies to do this (if the authors believe doing so is a good thing). I recommend publication of the paper regardless, but this comment is meant as a nudge to journals, editors, and authors to adopt open science practices whenever possible.

Our response:

We agree that the best practice is to publish the data. However, the data from the Raine Study cannot be made available in the public domain. Our data accessibility statement reads:

“The data were collected as part of the Raine Study. No data from the Raine Study can be made available in the public domain as the Raine Study is committed to a high level of confidentiality of the data in line with the informed consent provided by participants. The authors and the Raine Study will assist however to provide data access for verification purposes. Interested researchers can apply to access the data through the Raine Study processes.”

7. It is good that you pre-registered the study, but the pre-registration seems to have gaps. For example, I did not see where the pre-registration explains why you think the sample size of your study has appropriate power for testing these research questions, and it does not explain how exactly you plan to interpret the results of your statistical analyses given the low power of the study. For example, when a study is underpowered, I always tell my students to pay careful attention to effect size estimates for drawing conclusions not rely on p-values alone (since p-values by definition will be over .05 in a low-powered study even when there is a true effect). You may want to recommend practical steps for improving pre-registration (e.g. more transparency in power analysis and results interpretation) in future pre-registrations if you think it is feasible to do so. You could also recommend registered reports, which would allow colleagues to peer review study designs and analysis plans and provide recommendations for improvement.

Our response:

Ex-ante power calculations are especially helpful to determine what sample size is needed to detect a certain effect. In our case, this does not apply, as we simply tried to maximize our power, by recruiting as many participants as possible. We now mention this on page 8:

“We did not perform any ex-ante power calculations, as we aimed to maximize our power by inviting all active participants in the Raine Study.”

8. If there is space, the authors may be able to present additional limitations/directions for future work. For example, they did the study online and only paid some but not all participants based

on decisions. It is possible that this approach did not generate sufficient engagement of motivational, affective, and cognitive neural systems that sex hormones may modulate. Related to this point, a constraints on generality statement could potentially be useful to include (Simons et al., 2017).

Our response:

Thank you for this suggestion. We included a constraints on generality statement (Simons et al., 2017) in the discussion section on page 13. It reads:

“Our conclusions have several constraints on generality (cf. Simons et al. 2017). Our sample consists of young Western Australian adults. We believe that our findings would generalize to other adult samples, yet encourage continuous explorations of the effects, particularly in older adults and populations from non-Western societies. In terms of materials, we believe that our findings would generalize to other measurement methods that quantify economic preferences using incentivized economic games or validated survey questions, answered in private. Furthermore, we highlight the importance of measuring sex hormones at birth under the same conditions as stipulated in the paper. We have no reason to believe that the results depend on other characteristics of the participants, materials, or context.”

While we do think it is essential that studies use incentivized measures, we have no reason to believe that paying a random subsample a large amount affects results (compared to paying all participants a moderate amount). Paying a random subsample is frequently used in large-sample studies (e.g. Dohmen et al. 2011). Previous studies suggest that paying a random subsample a (large) amount, does not affect results compared to paying all subjects a (small) amount (e.g. Bolle 1990, Armentier 2006) and there is suggestive evidence that participants prefer the method we use (mentioned in the online appendix of Abdellaoui et al. (2011)). We now mention this on page 3 in the SM.

Thanks again to the authors for doing this important work. My comments are meant to help improve the paper further.

Our response:

Thanks for your guidance throughout the review process, we think this improved the paper substantially.